# Coupling Reactions of Anhydro-Aldose Tosylhydrazones with Boronic Acids

**DOI:** 10.3390/molecules27061795

**Published:** 2022-03-09

**Authors:** Tímea Kaszás, Balázs Áron Baráth, Bernadett Balázs, Tekla Blága, László Juhász, László Somsák, Marietta Tóth

**Affiliations:** Department of Organic Chemistry, University of Debrecen, P.O. Box 400, H-4002 Debrecen, Hungary; kaszas.timea@science.unideb.hu (T.K.); balazs.barath01@gmail.com (B.Á.B.); balazs.bernadetti@gmail.com (B.B.); tekla.tblaga@gmail.com (T.B.); juhasz.laszlo@science.unideb.hu (L.J.)

**Keywords:** coupling, anhydro-aldose tosylhydrazones, *C*-glycosides, heptenitols

## Abstract

A catalyst-free coupling reaction between *O*-peracetylated, *O*-perbenzoylated, *O*-permethylated, and *O*-permethoxymethylated 2,6-anhydro-aldose tosylhydrazones (*C*-(β-d-glycopyranosyl)formaldehyde tosylhydrazones) and aromatic boronic acids is reported. The base-promoted reaction is operationally simple and exhibits a broad substrate scope. The main products in most of the transformations were open-chain 1-*C*-aryl-hept-1-enitol type compounds while the expected β-d-glycopyranosylmethyl arenes (benzyl *C*-glycosides) were formed in subordinate yields only. A mechanistic rationale is provided to explain how a complex substrate may change the well-established course of the reaction.

## 1. Introduction

*N*-Tosylhydrazones have extensively been used in organic synthesis for more than half a century. In the past decade *N*-tosylhydrazones were generally applied in a variety of carbon–carbon and carbon–heteroatom bond forming reactions [1,2,3,4,5,6]. These transition metal catalyzed or catalyst-free cross-coupling reactions proceed through the in situ generated diazo compounds, followed by the formation of metal–carbene or carbene intermediates, which lead to the corresponding coupled products. Carbohydrate tosylhydrazones are also known, but their application in coupling reactions is poorly investigated.

In our research group an easy, one-step method was worked out for the synthesis of anhydro-aldose tosylhydrazones from readily accessible glycosyl cyanides [7,8,9]. We began a systematic study aimed at the investigation of the applicability of anhydro-aldose-tosylhydrazones **1** [7,8,9] in coupling reactions. In this project C-O [10], C-S [11], and C-N [12] bonds were successfully formed under metal-free conditions, while C-C bonds [13,14] were obtained in Pd-catalyzed reactions (Figure 1).

The metal-free reaction between the diazo precursor *N*-tosylhydrazones and alkyl, alkenyl, and arylboronic acids has been established in recent years as a powerful C(sp^3^)–C bond-forming transformation (Figure 2a) that avoids the application of precious metal catalysts and highly air/moisture-sensitive or expensive coupling partners [15,16]. However, this reaction was primarily limited to benzylic, α-heterocyclic, and/or aldehyde-derived tosylhydrazones at the substrate level, with lower yields observed for substrates that differed from these [15,17,18,19,20]. Dai and coworkers expanded this reductive coupling to acylferrocene tosylhydrazones, producing highly substituted α-arylalkylferrocenes [21]. *N*-Tosylhydrazones derived from 2-, 3-, and 4-substituted cyclohexanones and 4-substituted cyclopentanone were also used in couplings with alkenyl boronic acids [22]. The reductive coupling of *N*-tosylhydrazones under the standard reaction conditions was also examined with diarylborinic acids (Ar_2_B(OH)) to give diarylmethanes with good yields [23]. Kirschning developed a flow protocol for the reductive coupling reaction of *N*-tosylhydrazones with aryl boronic acids. To increase the practical applicability of the reaction, a two-step continuous flow protocol, starting with carbonyl compounds and tosylhydrazide, was also developed [24]. Nakagawa and coworkers expanded the scope of the transformation to a set of challenging heterocycle-containing aldehyde tosylhydrazones, such as those of protected azetidine, imidazole, and azaindole derivatives. These couplings resulted in low to good yields of drug-like molecules, bicyclic products, with a methylene linker between the rings (Figure 2b) [25]. This type of coupling of indole-3-carbaldehyde tosylhydrazone with boronic acids was used for the synthesis of biologically important 3-benzyl indole derivatives (Figure 2b) [26]. Ley and coworkers used the procedure for the metal-free coupling of 4-, 5-, and 6-membered saturated heterocyclic *p*-methoxyphenyl (PMP) sulfonylhydrazones with (het)aryl boronic acids to form sp^2^−sp^3^ linked bicyclic building blocks, including oxetanes, piperidines, and azetidines, from their parent ketones (Figure 2c) [27]. The reductive coupling was also applied for the synthesis of 9-arylfluorenes (Figure 2d) [28]. Thus, a wide range of 9-arylfluorenes was prepared in a one-pot process from 9-fluorenones by treatment with *N*-tosylhydrazide, followed by the reductive coupling of (het)aryl and alkyl boronic acids in the presence of potassium carbonate. A similar protocol was applied for the synthesis of triarylmethanes from less reactive diaryl ketones (Figure 2d) [29] and 1(or 2)-(1-phenylethyl)naphthalenes from acetyl naphthalene derivatives [30]. Wang and coworkers developed a three-component transition-metal-free reaction from α-halo-*N*-tosylhydrazones in the presence of *N*-alkylindoles and arylboronic acids to form a range of 3-substituted indoles [31]. A new type of cascade cyclization by reaction of alkenylboronic acids with 2-cyanoethyl or 3-cyanopropylcyclohexanone *N*-tosylhydrazones was developed by Valdés et al. [32,33]. A similar reaction between γ-azido-*N*-tosylhydrazones and boronic acids led to the formation of 2,2-disubstituted pyrrolidines in a domino process under microwave activation [34].

As the tosylhydrazone-boronic acid coupling can be of a great potential to avoid the utility of costly and poisonous metals and ligands, metal-free coupling reactions of boronic acids with anhydro-aldose tosylhydrazones were examined as a new type of substrate with higher complexity in comparison to the previous ones (Figure 2e). This transformation offers a simple possibility for the formation of *C*-glycosylmethyl derivatives whose preparation is rather cumbersome in the literature [13,35,36,37,38,39,40,41,42,43]. Herein we disclose our experiences with this reaction using various sugar configurations, protecting groups and boronic acids.

## 2. Results and Discussion

We started our study with the reaction between *O*-perbenzoylated *C*-(β-d-glucopyranosyl)formaldehyde tosylhydrazone **1a** [7,8,9] and phenylboronic acid (Table 1). First, the literature conditions [15] were applied using 1.5 equivalents of boronic acid and 1.5 equivalents of K_2_CO_3_ as the base in dry dioxane at reflux temperature (entry 1). The transformation resulted in a complex mixture, containing heptenitols **3a** and **4a** and *exo*-glucal **5** [8,44,45] but we did not observe the formation of the expected *C*-glucoside **2a** [13]. However, it can be assumed that the formation of the open chain compounds might occur by a base mediated ring-opening process, whose driving force could be the resonance stabilization of styrene **3a**. Similar heptenitols were obtained by the Wittig reaction [46,47]. Migration of a benzoyl protecting group could result in **4a**, and intramolecular carbene insertion into the C-2-H bond yielded *exo*-glucal **5** [8,44,45]. With other bases (Bu_4_NF, LiO*t*Bu, and K_3_PO_4_) the formation of the coupled product **2a** could also not be observed (entries 2–4). Instead, we obtained variable amounts of the heptenitols **3a** and **4a**, and *exo*-glucal **5**. Increasing the amount of K_3_PO_4_ raised the yield of heptenitol **3a** to 43% (entry 5). The effects of solvents other than dioxane were also studied, but in each case, complex reaction mixtures were obtained (entries 6–8). On the other hand, performing the reaction in the presence of five equivalents of phenylboronic acid with three or four equivalents of K_3_PO_4_ gave the *C*-glucoside **2a** in a very low yield beside **3a**, while **4a** and **5** were also isolated (entries 9 and 10). Raising the base excess gave *exo*-glucal **5** in moderate yield and heptenitols **3a** and **4a** in traces (entry 11). The best result was achieved with 20-fold excess of phenylboronic acid and 10-fold excess of K_3_PO_4_, to give heptenitol **3a** in 70% yield (entry 12). Thus, instead of the expected *C*-glycosylmethylarene derivative **2a**, an open chain compound, **3a**, proved to be the main product of the transformation.

To avoid base mediated side reactions, such as the acyl migration, *C*-(β-d-glucopyranosyl)formaldehyde tosylhydrazone Li-salt **1b** [10,12] was used for the couplings, where no added base is needed. Attempted reactions under UV irradiation (λ = 254 nm and 368 nm) carried out in a quartz tube proved to be totally ineffective, resulting in complex reaction mixtures. However, thermic conditions gave, generally, **3a** as the main product, besides *C*-glucoside **2a** and *exo*-glucal **5** (entries 13 and 14). Although the application of 10 equivalents of boronic acid significantly increased the yields (entry 15), the Li-salt reactions appeared less effective. Thus, tosylhydrazone **1a** and 1.5 or 20-fold excess of a boronic acid and 3 or 10-fold excesses of K_3_PO_4_ were used in further transformations.

The coupling reaction of **1a** was also examined with a variety of aryl boronic acids under the conditions selected above. These reactions resulted in varying yields of compound types **2**–**5**, among which the heptenitols **3** and **4** were the main products (Table 2). Application of higher excess of boronic acids and K_3_PO_4_ improved the yields in couplings with 4-(dibenzofuranyl) and 4-methoxyphenyl boronic acids (compare entries 3–4 and 6–7), but in other cases, this had no significant effect on the reaction outcome (compare entries 1–2, 10–11 and 12–13). The coupling was found to be significantly affected by the substituents on the aromatic ring; boronic acids with electron-releasing (entries 1–7) and chloro (entries 8 and 9)-substituents gave better yields. However, with the strong electron-withdrawing nitro group (entries 10–13) *exo*-glucal **5** was the main product, the coupled compound **2h** was observed in only one case. Isolation of the products in pure state often encountered difficulties. Due to very similar mobilities in silica gel column chromatography, *C*-glucosyl compounds **2** were polluted with the *exo*-glucal **5**, and heptenitols **3** and **4** polluted each other, therefore the yields were generally calculated on the basis of the ^1^H NMR spectra (Appendix A).

The coupling of *O*-peracetylated *C*-(β-d-galactopyranosyl)formaldehyde tosylhydrazone (**6**, Table 3) with phenylboronic acid was also investigated. With 1.5 equivalents of phenylboronic acid and 3 equivalents of potassium carbonate, only traces of the known compound types **7**, **8**, and **10** [8,44,45] were detected in the complex product mixture (entry 1), but with a 20-fold excess of the boronic acid *C*-(galactosyl)phenylmethane **7** was formed in low yield and heptenitols **8** and **9** proved to be the main products (entry 2). A compound with a free 6-OH (analogue of **3**), though might be formed, could not be detected possibly due to a faster acetyl migration to give **8** and **9**.

The NMR analysis provided evidence for the structure of all of the above derivatives and these are illustrated here by the examples of compounds **2**, **3**, and **4**. Anhydro-heptitol **2a**, synthesized in our group earlier [13], showed characteristic ^1^H NMR resonances for the C-1 methylene (δ 2.96 ppm (H-1_a_), 2.92 ppm (H-1_b_), with a great geminal coupling constant (12.3 Hz) between them) and the H-2 (‘anomeric’) protons (4.00 ppm). The characteristic ^13^C NMR resonances were δ 38.0 ppm (C-1) and 79.2 ppm (C-2). ^1^H and ^13^C NMR analysis of *C*-glycosyl derivatives **2b**,**c**,**f**,**h** showed similar chemical shifts for H-1a (2.92–3.44 ppm), H-1b (2.90–3.29 ppm), H-2 (3.98–4.33 ppm), C-1 (32.1–38.1), and C-2 (77.9–79.2) with geminal coupling constants of H-1_a_-1_b_ in the range of 14.3–15.0 Hz. These data indicated the similar structure of the *C*-glycosyl derivatives **2**. Ring-opened heptenitols **3** and **4** showed quite different spectral data. Signals characteristic for C-1 and C-2 of compounds **2** in the above ranges were missing in the ^13^C NMR spectra of **3** and **4**, instead resonances for –CH= type carbons in the ranges 130.8–136.9 ppm (for C-1) and 119.6–125.9 ppm (for C-2) appeared to prove the presence of a double bond in the molecules. The acyclic form was evidenced by the small vicinal coupling constants (in the range of 0.8–8.9 Hz). The great values (14.9–16.3 Hz) of coupling constant *^3^J*_1,2_ proved the *E*-configured double bond C-1=C-2 in these structures. The position of the free OH groups of heptenitols **3** and **4** were confirmed by observing cross peaks between OH and H-6 in heptenitols **3** and OH and H-5 in molecules **4** in their ^1^H–^1^H COSY spectra.

To further prove the formation of heptenitols and acyl group migration, benzoylation/acetylation of the corresponding compounds under standard conditions were carried out. Benzoylation [47] of the mixture of heptenitols **3** and **4** resulted in a single product **11** (Table 4) while acetylation [48] of heptenitol **9** gave *O*-peracetylated product **12** in good to excellent yields (Figure 3).

To get an insight into the effect of hydrolytically resistant ether type protecting groups on the outcome of the studied coupling reactions, *O*-permethylated (β-d-glucopyranosyl)formaldehyde tosylhydrazone **17** was synthesized. Methyl glucoside **13** was *O*-permethyled to get **14** [49] which was converted to the acetate derivative **15** [50] (Figure 4). On reacting **15** with trimethylsilyl cyanide in the presence of boron trifluoride etherate, cyanide **16** [51] was obtained. The anomers were separated by column chromatography. Then, β-cyanide **16β** was reduced in the presence of tosylhydrazide to give β-d-glucosyl tosylhydrazone **17** as a mixture of *E* and *Z* isomers.

Couplings with **17** gave cleaner product mixtures in better yields, and resulted in *C*-glucosides **18** (Table 5, entries 2, 4, and 8) or open-chain heptenitols **19** and **20** as the main products (entries 1, 5, 6, 7, 9, 10). *Exo*-glucal **21** [52] was always formed as a by-product. Compounds **18** and **21** proved inseparable, similar to open chain isomers **19** and **20**.

The transformation was extended to the acetal protected galactose derivative **24**, which was synthesized from the galactosyl cyanide **22** in two steps. Compound **22** was reacted with methoxymethyl chloride to obtain cyanide **23** [53], then a reduction step in the presence of tosylhydrazide gave a mixture of *E* and *Z* isomers of **24** (Figure 5).

The coupling reation of **24** with phenylboronic acid resulted in *E* heptenitol **26** as the main product and an inseparable mixture of *C*-(galactopyranosyl)phenylmethane **25** and *exo*-galactal **28** [53]. The *Z* isomer **27** was also detected in the mixture (Figure 6).

For the structure elucidation of Me (**18**–**20**) and MOM (**25**–**27**), protected derivatives 1D-NMR (^1^H, ^13^C) and 2D-NMR (^1^H–^1^H COSY, HSQC, and HMBC) spectra were recorded. The characteristic chemical shifts of C-1 (32.0–38.1 ppm vs. 132.3–134 ppm) and C-2 (79.0–80.9 ppm vs. 124.3–130.4 ppm) clearly revealed the structures of the anhydro-heptitols **18**, **25**, and heptenitols **19**, **20**, **26**, **27**, respectively.

In contrast to the transformations of acylated derivatives **2** and **7**, those of tosylhydrazones **17** and **24** possessing ether-type protecting groups (Me, MOM) resulted in no migration of the protecting groups as expected, but the *E* and *Z* isomers of the acyclic derivatives were isolated. The configuration of the double bonds was identified by the vicinal coupling constants being 16.0 Hz for the *E* and 11.4–12.1 Hz for the *Z* isomers. The measured vicinal coupling constants showed high variety for heptenitols **19** and **20**, in contrast to the cyclic ^4^*C*_1_ conformers **18**, where these values were 8.7 and 9.8 Hz for the trans diaxial protons. The position of the free OH groups of heptenitols **19**, **20**, **26**, and **27** were confirmed by observing cross peaks between OH and H-6 in their ^1^H–^1^H COSY spectra.

**Table 6 molecules-27-01795-t006:** Examination of possible ring opening of some anhydro-heptitols.

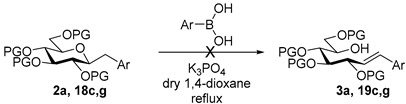							
**Entry**		**PG**	**Reaction Conditions**				**Experience**
			**Ar**	**Boronic Acid (Equiv.)**	**K_3_PO_4_** **(Equiv.)**	**t** **(h)**	
1	**2a**	Bz	Ph	-	10	22	partial deprotection
2	**18g**	Me	4-NO_2_C_6_H_4_	-	3	21	no conversion
3	**18g**	Me	4-NO_2_C_6_H_4_	1.5	-	21	no conversion
4	**18c**	Me	4-CF_3_C_6_H_4_	1.5	3	21	no conversion

**Table 7 molecules-27-01795-t007:** Examination of possible ring closing of heptenitols.

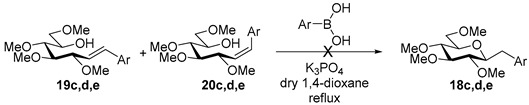
**Entry**		**Reaction Conditions**	**Experience**
		**Ar**	**Boronic Acid (Equiv.)**	**K_3_PO_4_** **(Equiv.)**	**t** **(h)**	
1	**19c, 20c**	4-CF_3_C_6_H_4_	1.5	3	21	no conversion
2	**19d, 20d**	4-FC_6_H_4_	-	3	21	no conversion
3	**19e, 20e**	4-ClC_6_H_4_	1.5	-	21	no conversion

To obtain more information about the formation of the open-chain heptenitols, first we checked the possibility of the ring opening of the anhydro-heptitols under the reaction conditions. Thus, **2a** was reacted with K_3_PO_4_ but partial deprotection of **2a** was observed only, without the formation of **3a** (Table 6, entry 1). The methyl protected derivatives **18c** or **18g** reacted neither in the presence of K_3_PO_4_, nor of a boronic acid or both (entries 2–4).

Next, formation of *C*-(glycosyl)arylmethane derivatives **18c**,**d**,**e** was examined from the corresponding heptenitols **19c**,**d**,**e** and **20c**,**d**,**e**. Attempted reactions in the presence of base and/or boronic acid resulted in no conversion (Table 7).

Based on these observations, it can be concluded that the cyclic *C*-glycosylmethyl derivatives and the open-chain heptenitols are not interconvertible under the applied conditions, they must be formed from the same intermediate during the reaction.

To explain these experiences, the following mechanistic possibilities can be considered (Figure 7). Loss of a sulfinate ion from tosylhydrazones **I** upon deprotonation or from Li-salt **V** may lead to the diazo intermediate **VI** which can give rise to carbene **VII** by eliminating a nitrogen molecule. The zwitterionic intermediate **VIII**, which arises from carbene **VII** (*path a*) or boronate complex **X**, formed from the diazo compound **VI** (*path b*), may lead to intermediate **IX**. Then, protodeboronation of **IX** under basic conditions can give anhydro-heptitol type products **III** (*path c*). Nevertheless, in intermediate **IX**, the ring oxygen, as a Lewis base, can attack the electron deficient boron atom to form the open chain heptenitol borate **XI** (*path e*) which, upon hydrolysis, can lead to the isolated heptenitols **IV**. The driving force of this rearrangement may be the conjugation of the double bond with the aromatic system, leading to an energetically more stable species. The standard by-product e*xo*-glycal **II** can be formed by an intramolecular insertion reaction of carbene **VII** (*path d*).

## 3. Conclusions

This study on the metal-free coupling reactions of *C*-(β-d-glycopyranosyl)formaldehyde (2,6-anhydro-aldose) tosylhydrazones with aromatic boronic acids revealed that the main reaction pathway was the formation of ring-opened hept-1-enitol derivatives, while the expected *C*-glycopyranosyl compounds (benzyl *C*-glycosides) were formed only in low to moderate yields. The corresponding *exo*-glycals always appeared as unavoidable by-products. *O*-Acyl protecting groups on the carbohydrate moieties underwent migrations which further increased the number of products in the otherwise rather complex reaction mixtures. Tosylhydrazones with ether type *O*-protections gave cleaner reactions but resulted in the same product types in similar ratios. The suggested mechanistic rationale explained how the complex sugar-derived tosylhydrazone substrates changed the reaction pathway. We think that this study also highlights the importance of transformations of high complexity which, though resulting in several products, may lead to a better understanding of their mechanism and may thus inspire further work.

## 4. Experimental

### 4.1. General Methods

Optical rotations were determined with a Perkin–Elmer 241 polarimeter or Jasco P-2000 (Easton, MD, USA) at room temperature. NMR spectra were recorded with a Bruker AM Avance DRX 360 MHz (360/90 MHz for ^1^H/^13^C) or Bruker AM Avance I 400 MHz (400/100 MHz for ^1^H/^13^C) or Bruker AM Avance II 500 MHz (500/125 MHz for ^1^H/^13^C) spectrometers. Chemical shifts are referenced to TMS as the internal reference (^1^H), or to the residual solvent signals (^13^C). The assignments of the ^1^H and ^13^C NMR signals of compounds **2**–**4**, **7**–**9**, **11**, **12**, **18**–**20**, and **25**–**27** were performed by their COSY (**2a**, **3a**,**c**, **4a**,**e**, **7**, **8**, **9**, **11a**,**b**, **12**, **18b**,**f**,**i**, **19a**,**c**,**h**,**i**, **20a**,**d**,**i**, **25**, **26**, **27**), HSQC (**2a**, **3a**,**c**, **4a**,**e**, **7**, **8**, **9**, **11a**,**b**, **12**, **18b**,**f**,**i**, **19a**,**c**,**h**,**i**, **20a**,**d**,**i**, **25**, **26**, **27**), or HMBC (**3a**,**c**, **4a**,**e**, **7**, **8**, **9**, **11a**,**b**, **12**, **18b**,**e**,**f**,**i**, **19a**,**c**,**h**,**i**, **20a**,**d**,**i**, **25**, **26**, **27**) spectra. Mass spectra were recorded with maXis II UHR ESI-QTOF MS (Bruker Daltonik, Bremen, Germany) instruments in positive ion mode with the electrospray ionization technique, or Thermo LTQ XL (Thermo Electron Corp., San Jose, CA, USA) mass spectrometers operated in a full scan positive ion ESI and APCI mode. TLC was performed on a DCAlurolle Kieselgel 60 F254 (Merck). TLC plates were visualized under UV light, and by gentle heating (generally no spray reagent was used but, if more intense charring was necessary, the plate was sprayed with the following solution: abs. EtOH (95 mL), cc. H_2_SO_4_ (5 mL), anisaldehyde (1 mL)). For column chromatography Kieselgel 60 (Merck, particle size (0.063–0.200 mm) was applied. The compound 1,4-dioxane was distilled from sodium benzophenone ketyl and stored over sodium wires.

### 4.2. General Procedure I: Conditions for the Reaction of Anhydro-Aldose Tosylhydrazones with Boronic Acids

A boronic acid (1.5 or 20 mmol, specified with the particular reactions) and K_3_PO_4_ (3 or 10 mmol, specified with the particular reactions) were suspended in dry 1,4-dioxane (15 mL). The suspension was stirred and heated to reflux, and then a solution of a tosylhydrazone (**1**; **17** or **24**, 1 mmol) in dry 1,4-dioxane (15 mL) was added dropwise over ~20 min. When TLC (1:2 EtOAc–hexane for **1** and **17**, 1:1 EtOAc–hexane for **24**) indicated complete consumption of the starting compound (20 min–4 h), the mixture was cooled down and the insoluble material was filtered off and washed thoroughly with dry 1,4-dioxane (3 × 20 mL). The solvent was removed under reduced pressure, and the residue was purified by column chromatography, with eluents indicated for the particular compounds to give anhydro heptitols and hept-1-enitols.

### 4.3. Characterization of Anhydro-Heptitols ***2***

#### 4.3.1. 2,6-Anhydro-3,4,5,7-Tetra-*O*-Benzoyl-1-Deoxy-1-Phenyl-d-*glycero*-d-*gulo*-Heptitol (**2a**)

Isolated from a reaction of tosylhydrazone **1a** (0.10 g, 0.13 mmol), phenylboronic acid (1.5 equiv., 0.02 g, 0.19 mmol), and K_3_PO_4_ (3 equiv., 0.08 g, 0.39 mmol) according to General procedure I by column chromatography (1:2 EtOAc–hexane) to yield 3 mg (4%) of **2a** as a white amorphous product. Optical rotation, NMR and MS spectra are identical with those reported [13].



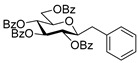



#### 4.3.2. 2,6-Anhydro-3,4,5,7-Tetra-*O*-Benzoyl-1-Deoxy-1-(Naphth-2-yl)-d-*glycero*-d-*gulo*-Heptitol (**2b**)

Isolated from a reaction of tosylhydrazone **1a** (0.10 g, 0.13 mmol), naphthalen-2-ylboronic acid (20 equiv., 0.44 g, 2.57 mmol), and K_3_PO_4_ (10 equiv., 0.27 g, 1.29 mmol) according to General procedure I by column chromatography (1:4 EtOAc–hexane) to yield 4 mg (4%) of **2b** as a pale brown amorphous solid. R_f_: 0.42 (1:2 EtOAc–hexane); [α]_D_ + 6 (*c* 0.16, CH_2_Cl_2_). ^1^H NMR (400 MHz, CDCl_3_) δ 7.99–7.19 (27H, m, aromatics), 5.90 (1H, pszeudo t, *J*_4,5_ 9.6 Hz, H-4), 5.62 (1H, pseudo t, *J*_5,6_ 9.7 Hz, H-5), 5.52 (1H, pseudo t, *J*_3,4_ 9.6 Hz, H-3), 4.57 (1H, dd, *J*_7a,7b_ 12.0 Hz, H-7_a_), 4.41 (1H, dd, H-7_b_), 4.09 (1H, ddd, *J*_1a,2_ 5.1, *J*_1b,2_ 6.6, *J*_2,3_ 9.8 Hz, H-2), 4.04 (1H, ddd, *J*_6,7a_ 2.7, *J*_6,7b_ 6.3 Hz, H-6), 3.12 (1H, dd, *J*_1a,1b_ 14.8 Hz, H-1_a_), 3.08 (1H, dd, H-1_b_). ^13^C NMR (90 MHz, CDCl_3_) δ 166.3, 166.1, 165.6, 165.5 (4 × CO), 136.6–124.7 (aromatics), 79.2 (C-2), 76.3 (C-6), 74.7 (C-4), 72.6 (C-3), 70.1 (C-5), 63.6 (C-7), 38.3 (C-1). HR-ESI-MS positive mode (m/z): calc. for [M + Na]^+^ = 743.2252, found: [M + Na]^+^ = 743.2253; C_45_H_36_O_9_ (720.24).



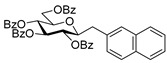



#### 4.3.3. 2,6-Anhydro-3,4,5,7-Tetra-*O*-Benzoyl-1-(4-Dibenzo[b,d]furanyl)-1-Deoxy-d-*glycero*-d-*gulo*-Heptitol (**2c**)

Isolated from a reaction of tosylhydrazone **1a** (0.10 g, 0.13 mmol), dibenzo[*b,d*]furan-4-ylboronic acid (20 equiv., 0.55 g, 2.57 mmol), and K_3_PO_4_ (10 equiv., 0.27 g, 1.29 mmol) according to General procedure I by column chromatography (1:3 EtOAc–hexane) to yield 30 mg pale brown amorphous solid containing **2c** and **5** in 1:1 ratio. R_f_: 0.50 (1:2 EtOAc–hexane). ^1^H NMR (400 MHz, CDCl_3_) δ 8.21–6.93 (27H, m, aromatics), 5.91 (1H, pseudo t, *J*_4,5_ 9.5 Hz, H-4), 5.64 (1H, pseudo t, *J*_5,6_ 9.8 Hz, H-5), 5.52 (1H, pseudo t, *J*_3,4_ 9.8 Hz, H-3), 4.56 (1H, dd, *J*_7a,7b_ 12.0 Hz, H-7_a_), 4.42 (1H, dd, H-7_b_), 4.33 (1H, ddd, *J*_1a,2_ 3.2, *J*_1b,2_ 8.0, *J*_2,3_ 9.8 Hz, H-2), 4.07 (1H, ddd, *J*_6,7a_ 2.9, *J*_6,7b_ 5.9 Hz, H-6), 3.44 (1H, dd, *J*_1a,1b_ 14.6 Hz, H-1_a_), 3.29 (1H, dd, H-1_b_). ^13^C NMR (100 MHz, CDCl_3_) δ 166.3, 166.1, 165.5 (4 × CO), 135.6–110.4 (aromatics), 77.9 (C-2), 76.2 (C-6), 74.8 (C-4), 72.5 (C-3), 70.1 (C-5), 63.5 (C-7), 32.1 (C-1). HR-ESI-MS positive mode (m/z): calc. for [M + H]^+^ = 761.2381, found: [M + H]^+^ = 761.2379; C_47_H_36_O_10_ (760.23).



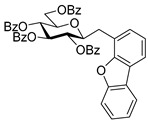



#### 4.3.4. 2,6-Anhydro-3,4,5,7-Tetra-*O*-Benzoyl-1-(3-Chlorophenyl)-1-Deoxy-d-*glycero*-d-*gulo*-Heptitol (**2f**)

Isolated from a reaction of tosylhydrazone **1a** (0.10 g, 0.13 mmol), 3-chlorophenylboronic acid (1.5 equiv., 0.03 g, 0.19 mmol), and K_3_PO_4_ (3 equiv., 0.08 g, 0.39 mmol) according to General procedure I by column chromatography (1:4 EtOAc–hexane) to yield 13 mg white amorphous solid containing **2f** and **5** in 1:2 ratio. R_f_: 0.48 (1:2 EtOAc–hexane). ^1^H NMR (400 MHz, CDCl_3_) δ 8.13–7.76 (12H, m, aromatics), 7.63–6.94 (12H, m, aromatics), 5.89 (1H, pseudo t, *J*_4,5_ 9.7 Hz, H-4), 5.60 (1H, pseudo t, *J*_5,6_ 9.7 Hz, H-5), 5.45 (1H, pseudo t, *J*_3,4_ 9.5 Hz, H-3), 4.57 (1H, dd, *J*_7a,7b_ 12.1 Hz, H-7_a_), 4.42 (1H, dd, H-7_b_), 4.05 (1H, ddd, *J*_6,7a_ 2.8, *J*_6,7b_ 6.2 Hz, H-6), 3.98 (1H, ddd, *J*_1a,2_ 5.3, *J*_1b,2_ 6.6, *J*_2,3_ 9.7 Hz, H-2), 2.92 (1H, dd, *J*_1a,1b_ 15.0 Hz, H-1_a_), 2.90 (1H, dd, H-1_b_). ^13^C NMR (90 MHz, CDCl_3_) δ 166.3, 166.1, 165.7, 165.6 (4 × CO), 156.3–125.7 (aromatics), 78.8 (C-2), 76.4 (C-6), 74.6 (C-4), 72.6 (C-3), 70.1 (C-5), 63.5 (C-7), 37.4 (C-1). HR-ESI-MS positive mode (m/z): calc. for [M + Na]^+^ = 727.1705, found: [M + Na]^+^ = 727.1708; C_41_H_33_ClO_9_ (704.18).



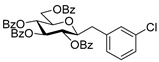



#### 4.3.5. 2,6-Anhydro-3,4,5,7-Tetra-*O*-Benzoyl-1-Deoxy-1-(4-Nitrophenyl)-d-*glycero*-d-*gulo*-Heptitol (**2h**) 

Isolated from a reaction of tosylhydrazone **1a** (0.30 g, 0.39 mmol), 4-nitrophenylboronic acid (20 equiv., 1.30 g, 7.72 mmol), and K_3_PO_4_ (10 equiv., 0.82 g, 3.86 mmol) according to General procedure I by column chromatography (1:3 EtOAc–hexane) to yield 32 mg pale brown amorphous solid containing **2h** and **5** in 4:1 ratio. R_f_: 0.44 (1:2 EtOAc–hexane). ^1^H NMR (400 MHz, CDCl_3_) δ 8.32–7.72 (8H, m, aromatics), 7.69–7.16 (16H, m, aromatics), 5.90 (1H, pseudo t, *J*_4,5_ 9.5 Hz, H-4), 5.58 (1H, pseudo t, *J*_5,6_ 9.8 Hz, H-5), 5.45 (1H, pseudo t, *J*_3,4_ 9.7 Hz, H-3), 4.53 (1H, dd, *J*_7a,7b_ 12.2 Hz, H-7_a_), 4.48 (1H, dd, H-7_b_), 4.05 (1H, ddd, *J*_6,7a_ 3.2, *J*_6,7b_ 6.6 Hz, H-6), 3.99 (1H, ddd, *J*_1a,2_ 5.1, *J*_1b,2_ 7.0, *J*_2,3_ 9.7 Hz, H-2), 3.03 (1H, dd, *J*_1a,1b_ 14.3 Hz, H-1_a_), 3.02 (1H, dd, H-1_b_). ^13^C NMR (90 MHz, CDCl_3_) δ 166.4, 166.2, 165.7, 165.6 (4 × CO), 161.7–115.1 (aromatics), 78.3 (C-2), 76.4 (C-6), 74.5 (C-4), 72.5 (C-3), 70.0 (C-5), 63.3 (C-7), 37.8 (C-1). HR-ESI-MS positive mode (m/z): calc. for [M + Na]^+^ = 738.1946, found: [M + Na]^+^ = 738.1950; C_41_H_33_NO_11_ (715.21).



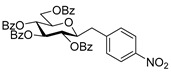



### 4.4. Characterization of Hept-1-Enitols ***3*** and ***4***

#### 4.4.1. (*E*)-3,4,5,7-Tetra-*O*-Benzoyl-1,2-Dideoxy-1-Phenyl-d-*gluco*-Hept-1-Enitol (**3a**) 

Prepared from tosylhydrazone **1a** (0.80 g, 1.03 mmol), phenylboronic acid (20 equiv., 2.51 g, 20.60 mmol), and K_3_PO_4_ (10 equiv., 2.19 g, 10.30 mmol) according to General procedure I. Purified by column chromatography (1:4 EtOAc–hexane) to yield 484 mg (70%) of **3a** as a white amorphous solid. R_f_: 0.36 (1:2 EtOAc–hexane); [α]_D_ + 21 (*c* 0.20, CH_2_Cl_2_). ^1^H NMR (500 MHz, CDCl_3_) δ 8.18–7.82 (8H, m, aromatics), 7.64–7.15 (17H, m, aromatics), 6.78 (1H, d, *J*_1,2_ 15.9 Hz, H-1), 6.32 (1H, dd, *J*_2,3_ 6.9 Hz, H-2), 6.14–6.02 (2H, m, H-3, H-4), 5.76 (1H, dd, *J*_4,5_ 0.8, *J*_5,6_ 8.9 Hz, H-5), 4.53 (1H, dd, *J*_6,7a_ 2.6, *J*_7a,7b_ 11.9 Hz, H-7_a_), 4.34 (1H, dd, *J*_6,7b_ 5.7 Hz, H-7_b_), 4.21–4.11 (1H, m, H-6), 3.58 (1H, d, *J*_6,OH_ 4.3 Hz, OH). ^13^C NMR (125 MHz, CDCl_3_) δ 167.3, 166.7, 165.6, 165.4 (4 × CO), 136.7 (C-1), 136.3–125.9 (aromatics), 122.1 (C-2), 73.9 (C-3), 73.3 (C-4), 71.3 (C-5), 68.6 (C-6), 65.5 (C-7). HR-ESI-MS positive mode (m/z): calc. for [M + Na]^+^ = 693.2095, found: [M + Na]^+^ = 693.2095; C_41_H_34_O_9_ (670.22).



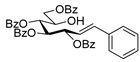



#### 4.4.2. (*E*)-3,4,6,7-Tetra-*O*-Benzoyl-1,2-Dideoxy-1-Phenyl-d-*gluco*-Hept-1-Enitol (**4a**)

Isolated from a reaction of tosylhydrazone **1a** (0.30 g, 0.39 mmol), phenylboronic acid (20 equiv., 0.94 g, 7.72 mmol), and K_3_PO_4_ (10 equiv., 8.20 g, 3.86 mmol) according to General procedure I by column chromatography (1:2 EtOAc–hexane) to yield 24 mg yellow amorphous solid containing **4a** and **3a** in 10:2 ratio. R_f_: 0.37 (1:2 EtOAc–hexane). ^1^H NMR (400 MHz, CDCl_3_) δ 8.22–7.77 (8H, m, aromatics), 7.63–7.06 (17H, m, aromatics), 6.99 (1H, d, *J*_1,2_ 15.6 Hz, H-1), 6.31 (1H, dd, *J*_2,3_ 8.0 Hz, H-2), 6.23 (1H, pseudo t, *J*_3,4_ 8.6 Hz, H-3), 5.82 (1H, dd, *J*_4,5_ 1.3 Hz, H-4), 5.44 (1H, ddd, *J*_6,7a_ 3.3, *J*_6,7b_ 4.4, *J*_5,6_ 8.0 Hz, H-6), 4.81 (1H, dd, *J*_7a,7b_ 12.4 Hz, H-7_a_), 4.74 (1H, dd, H-7_b_), 4.39 (1H, pseudo t, H-5), 3.25 (1H, d, *J*_5,OH_ 8.4 Hz, OH). ^13^C NMR (100 MHz, CDCl_3_) δ 167.3, 166.7, 165.7, 165.4 (4 × CO), 136.9 (C-1), 136.3–124.1 (aromatics), 122.7 (C-2), 74.6 (C-3), 72.4 (C-4), 71.7 (C-6), 68.5 (C-5), 63.4 (C-7). HR-ESI-MS positive mode (m/z): calc. for [M + Na]^+^ = 693.2095, found: [M + Na]^+^ = 693.2096; C_41_H_34_O_9_ (670.22).



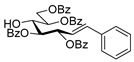



#### 4.4.3. (*E*)-3,4,5,7-Tetra-*O*-Benzoyl-1,2-Dideoxy-1-Naphth-2-yl-d-*gluco*-Hept-1-Enitol (**3b**) and (*E*)-3,4,6,7-Tetra-*O*-Benzoyl-1,2-Dideoxy-1-Naphth-2-yl-d-*gluco*-Hept-1-Enitol (**4b**)

Isolated from a reaction of tosylhydrazone **1a** (0.10 g, 0.13 mmol), naphthalen-2-ylboronic acid (20 equiv., 0.44 g, 2.57 mmol), and K_3_PO_4_ (10 equiv., 0.27 g, 1.29 mmol) according to General procedure I by column chromatography (1:4 EtOAc–hexane) to yield 70 mg pale brown amorphous solid containing **3b** and **4b** in 1.5:1 ratio. R_f_: 0.25 (1:2 EtOAc–hexane).



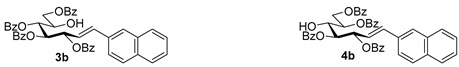



**3b**: ^1^H NMR (400 MHz, CDCl_3_) δ 8.20–7.03 (27H, m, aromatics), 6.94 (1H, d, *J*_1,2_ 15.9 Hz, H-1), 6.45 (1H, dd, *J*_2,3_ 6.7 Hz, H-2), 6.19–6.09 (2H, m, H-3, H-4), 5.82 (1H, dd, *J*_4,5_ 1.2, *J*_5,6_ 8.9 Hz, H-5), 4.54 (1H, dd, *J*_6,7a_ 3.0, *J*_7a,7b_ 11.9 Hz, H-7_a_), 4.35 (1H, dd, *J*_6,7b_ 5.7 Hz, H-7_b_), 4.24–4.13 (1H, m, H-6), 3.66 (1H, bs, OH). ^13^C NMR (100 MHz, CDCl_3_) δ 167.2, 166.7, 165.6, 165.4 (4 × CO), 136.6 (C-1), 136.4–123.3 (aromatics), 122.4 (C-2), 73.9 (C-3), 73.2 (C-4), 71.3 (C-5), 68.5 (C-6), 65.5 (C-7). HR-ESI-MS positive mode (m/z): calc. for [M + Na]^+^ = 743.2252, found: [M + Na]^+^ = 743.2250; C_45_H_36_O_9_ (720.24).

**4b**: ^1^H NMR (400 MHz, CDCl_3_) δ 8.20–7.03 (28H, m, aromatics, H-1), 6.44 (1H, dd, *J*_1,2_ 15.8, *J*_2,3_ 8.4 Hz, H-2), 6.31 (1H, pseudo t, *J*_3,4_ 8.9 Hz, H-3), 5.88 (1H, dd, *J*_4,5_ 1.4 Hz, H-4), 5.48 (1H, ddd, *J*_6,7a_ 3.3, *J*_6,7b_ 4.4, *J*_5,6_ 8.0 Hz, H-6), 4.82 (1H, dd, *J*_7a,7b_ 12.4 Hz, H-7_a_), 4.75 (1H, dd, H-7_b_), 4.44 (1H, d, H-5), 3.57 (1H, bs, OH). ^13^C NMR (100 MHz, CDCl_3_) δ 166.9, 166.3, 165.8, 165.4 (4 × CO), 136.9 (C-1), 136.4–123.0 (aromatics), 123.1 (C-2), 74.8 (C-3), 72.5 (C-4), 71.6 (C-6), 68.4 (C-5), 63.4 (C-7). HR-ESI-MS positive mode (m/z): calc. for [M + Na]^+^ = 743.2252, found: [M + Na]^+^ = 743.2254; C_45_H_36_O_9_ (720.24).

#### 4.4.4. (*E*)-3,4,5,7-Tetra-*O*-Benzoyl-1-(4-Dibenzo[b,d]furanyl)-1,2-Dideoxy-d-*gluco*-Hept-1-Enitol (**3c**)

Prepared from tosylhydrazone **1a** (0.10 g, 0.13 mmol), dibenzo[*b,d*]furan-4-ylboronic acid (1.5 equiv., 0.04 g, 0.19 mmol), and K_3_PO_4_ (3 equiv., 0.08 g, 0.39 mmol) according to General procedure I. Purified by column chromatography (1:2 EtOAc–hexane) to yield 16 mg (16%) of **3c** as a pale brown amorphous solid. R_f_: 0.32 (1:2 EtOAc–hexane); [α]_D_ + 5 (*c* 0.11, CH_2_Cl_2_). ^1^H NMR (400 MHz, CDCl_3_) δ 8.38–7.68 (12H, m, aromatics), 7.64–7.16 (15H, m, aromatics), 7.15–6.92 (2H, m, H-1, H-2), 6.22 (1H, dd, *J*_2,3_ 5.5, *J*_3,4_ 8.0 Hz, H-3), 6.16 (1H, dd, *J*_4,5_ 1.7 Hz, H-4), 5.87 (1H, dd, *J*_5,6_ 8.9 Hz, H-5), 4.54 (1H, dd, *J*_6,7a_ 3.0, *J*_7a,7b_ 11.9 Hz, H-7_a_), 4.35 (1H, dd, *J*_6,7b_ 5.7 Hz, H-7_b_), 4.23–4.15 (1H, m, H-6), 3.60 (1H, d, *J*_6,OH_ 5.3 Hz, OH). ^13^C NMR (100 MHz, CDCl_3_) δ 167.2, 166.7, 165.6, 165.4 (4 × CO), 130.8 (C-1), 156.5–111.9 (aromatics), 125.9 (C-2), 74.1 (C-3), 73.3 (C-4), 71.3 (C-5), 68.6 (C-6), 65.5 (C-7). C_47_H_36_O_10_ (760.23). HR-ESI-MS positive mode (m/z): calc. for [M + Na]^+^ = 783.2201, found: [M + Na]^+^ = 783.2202; C_47_H_36_O_10_ (760.23).



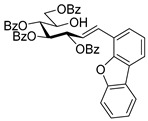



#### 4.4.5. (*E*)-3,4,6,7-Tetra-*O*-Benzoyl-1-(4-Dibenzo[b,d]furanyl)-1,2-Dideoxy-d-*gluco*-Hept-1-Enitol (**4c**)

Prepared from tosylhydrazone **1a** (0.10 g, 0.13 mmol), dibenzo[*b,d*]furan-4-ylboronic acid (20 equiv., 0.55 g, 2.57 mmol), and K_3_PO_4_ (10 equiv., 0.27 g, 1.29 mmol) according to General procedure I. Purified by column chromatography (1:3 EtOAc–hexane) to yield 29 mg (30%) of **4c** as a yellow amorphous solid. R_f_: 0.32 (1:2 EtOAc–hexane); [α]_D_ + 5 (*c* 0.11, CH_2_Cl_2_). ^1^H NMR (400 MHz, CDCl_3_) δ 8.23–6.76 (27H, m, aromatics), 7.02 (1H, d, *J*_1,2_ 16.2 Hz, H-1), 6.97 (1H, dd, *J*_2,3_ 8.2 Hz, H-2), 6.29 (1H, pseudo t, *J*_3,4_ 9.0 Hz, H-3), 5.91 (1H, dd, *J*_4,5_ 1.5 Hz, H-4), 5.44 (1H, ddd, *J*_6,7a_ 3.5, *J*_6,7b_ 4.4, *J*_5,6_ 8.0 Hz, H-6), 4.77 (1H, dd, *J*_7a,7b_ 12.4 Hz, H-7_a_), 4.68 (1H, dd, H-7_b_), 4.50–4.41 (1H, m, H-5), 3.28 (1H, d, *J*_5,OH_ 6.0 Hz, OH). ^13^C NMR (100 MHz, CDCl_3_) δ 167.1, 166.9, 165.9, 165.8 (4 × CO), 131.7 (C-1), 156.3–111.0 (aromatics), 120.6 (C-2), 75.1 (C-3), 72.5 (C-4), 71.7 (C-6), 68.4 (C-5), 63.3 (C-7). HR-ESI-MS positive mode (m/z): calc. for [M + Na]^+^ = 783.2201, found: [M + Na]^+^ = 783.2202; C_47_H_36_O_10_ (760.23).



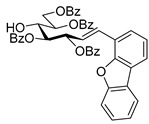



#### 4.4.6. (*E*)-3,4,5,7-Tetra-*O*-Benzoyl-1,2-Dideoxy-1-(4-Methylphenyl)-d-*gluco*-Hept-1-Enitol (**3d**) and (*E*)-3,4,6,7-Tetra-*O*-Benzoyl-1,2-Dideoxy-1-(4-Methylphenyl)-d-*gluco*-Hept-1-enitol (**4d**)

Isolated from a reaction of tosylhydrazone **1a** (0.10 g, 0.13 mmol), 4-methylphenylboronic acid (1.5 equiv., 0.03 g, 0.19 mmol), and K_3_PO_4_ (3 equiv., 0.08 g, 0.39 mmol) according to General procedure I by column chromatography (1:2 EtOAc–hexane) to yield 44 mg pale yellow amorphous solid containing **3d** and **4d** in 2:1 ratio. R_f_: 0.38 (1:2 EtOAc–hexane). 



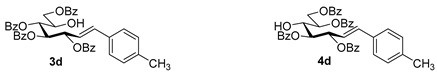



**3d**: ^1^H NMR (400 MHz, CDCl_3_) δ 8.20–7.81 (8H, m, aromatics), 7.64–7.01 (16H, m, aromatics), 6.74 (1H, d, *J*_1,2_ 15.9 Hz, H-1), 6.26 (1H, dd, *J*_2,3_ 6.7 Hz, H-2), 6.12–6.02 (2H, m, H-3, H-4), 5.75 (1H, dd, *J*_4,5_ 1.1, *J*_5,6_ 8.9 Hz, H-5), 4.52 (1H, dd, *J*_6,7a_ 2.9, *J*_7a,7b_ 11.9 Hz, H-7_a_), 4.34 (1H, dd, *J*_6,7b_ 5.9 Hz, H-7_b_), 4.20–4.10 (1H, m, H-6), 3.60 (1H, d, *J*_6,OH_ 5.2 Hz, OH), 2.33 (3H, s, CH_3_). ^13^C NMR (100 MHz, CDCl_3_) δ 167.3, 166.7, 165.6, 165.4 (4 × CO), 136.8 (C-1), 139.5–126.1 (aromatics), 120.9 (C-2), 74.0 (C-3), 73.4 (C-4), 71.4 (C-5), 68.6 (C-6), 65.5 (C-7), 21.4 (CH_3_). HR-ESI-MS positive mode (m/z): calc. for [M + Na]^+^ = 707.2252, found: [M + Na]^+^ = 707.2251; C_42_H_36_O_9_ (684.24).

**4d:** ^1^H NMR (400 MHz, CDCl_3_) δ 8.21–7.73 (8H, m, aromatics), 7.72–7.01 (16H, m, aromatics), 6.96 (1H, d, *J*_1,2_ 14.9 Hz, H-1), 6.25 (1H, dd, *J*_2,3_ 6.6 Hz, H-2), 6.21 (1H, pseudo t, *J*_3,4_ 8.6 Hz, H-3), 5.81 (1H, dd, *J*_4,5_ 1.3 Hz, H-4), 5.44 (1H, ddd, *J*_6,7a_ 3.4, *J*_6,7b_ 4.5, *J*_5,6_ 8.5 Hz, H-6), 4.79 (1H, dd, *J*_7a,7b_ 12.4 Hz, H-7_a_), 4.74 (1H, dd, H-7_b_), 4.39 (1H, pseudo t, *J*_5,OH_ 8.8 Hz, H-5), 3.16 (1H, bs, OH), 2.35 (3H, s, CH_3_). ^13^C NMR (100 MHz, CDCl_3_) δ 167.1, 166.9, 165.9, 165.3 (4 × CO), 135.9 (C-1), 139.5–115.0 (aromatics), 121.6 (C-2), 74.7 (C-3), 72.5 (C-4), 71.7 (C-6), 68.5 (C-5), 63.4 (C-7), 21.4 (CH_3_). HR-ESI-MS positive mode (m/z): calc. for [M + Na]^+^ = 707.2252, found: [M + Na]^+^ = 707.2254; C_42_H_36_O_9_ (684.24).

#### 4.4.7. (*E*)-3,4,5,7-Tetra-*O*-Benzoyl-1,2-Dideoxy-1-(4-Methoxyphenyl)-d-*gluco*-Hept-1-Enitol (**3e**) and (*E*)-3,4,6,7-Tetra-*O*-Benzoyl-1,2-Dideoxy-1-(4-Methoxyphenyl)-d-*gluco*-Hept-1-Enitol (**4e**)

Isolated from a reaction of tosylhydrazone **1a** (0.10 g, 0.13 mmol), 4-methoxyphenylboronic acid (1.5 equiv., 0.03 g, 0.19 mmol), and K_3_PO_4_ (3 equiv., 0.08 g, 0.39 mmol) according to General procedure I by column chromatography (1:2 EtOAc–hexane) to yield 43 mg yellow amorphous solid containing **3e** and **4e** in 2.5:1 ratio. R_f_: 0.31 (1:2 EtOAc–hexane). 



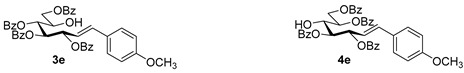



**3e**: ^1^H NMR (400 MHz, CDCl_3_) δ 8.28–7.67 (8H, m, aromatics), 7.65–7.09 (14H, m, aromatics), 6.81 (2H, d, *J* 8.8 Hz, aromatics), 6.71 (1H, d, *J*_1,2_ 15.7 Hz, H-1), 6.25–6.11 (1H, m, H-2), 6.10–6.02 (2H, m, H-3, H-4), 5.75 (1H, dd, *J*_4,5_ 1.0, *J*_5,6_ 8.9 Hz, H-5), 4.52 (1H, dd, *J*_6,7a_ 3.0, *J*_7a,7b_ 11.9 Hz, H-7_a_), 4.34 (1H, dd, *J*_6,7b_ 5.9 Hz, H-7_b_), 4.21–4.09 (1H, m, H-6), 3.80 (3H, s, OCH_3_), 3.63 (1H, bs, OH). ^13^C NMR (100 MHz, CDCl_3_) δ 167.3, 166.7, 165.6, 165.4 (4 × CO), 136.5 (C-1), 160.9–112.7 (aromatics), 119.6 (C-2), 74.2 (C-3), 73.4 (C-4), 71.4 (C-5), 68.5 (C-6), 65.5 (C-7), 55.4 (OCH_3_). HR-ESI-MS positive mode (m/z): calc. for [M + Na]^+^ = 723.2201, found: [M + Na]^+^ = 723.2204; C_42_H_36_O_10_ (700.23).

**4e**: ^1^H NMR (400 MHz, CD_3_OD) δ 8.19–7.74 (8H, m, aromatics), 7.65–7.12 (14H, m, aromatics), 6.96 (1H, d, *J*_1,2_ 15.8 Hz, H-1), 6.90 (2H, d, *J* 8.7 Hz, aromatics), 6.32 (1H, dd, *J*_2,3_ 8.2 Hz, H-2), 6.21 (1H, pseudo t, *J*_3,4_ 9.1 Hz, H-3), 5.82 (1H, dd, *J*_4,5_ 1.5 Hz, H-4), 5.45 (1H, ddd, *J*_6,7a_ 2.5, *J*_6,7b_ 5.1 Hz, H-6), 4.93 (1H, dd, *J*_7a,7b_ 12.2 Hz, H-7_a_), 4.57 (1H, dd, H-7_b_), 4.52 (1H, dd, *J*_5,6_ 9.1 Hz, H-5), 3.80 (3H, s, OCH_3_), 3.58 (1H, bs, OH). ^13^C NMR (90 MHz, CDCl_3_) δ 167.0, 166.9, 165.8, 165.4 (4 × CO), 136.6 (C-1), 160.9–112.7 (aromatics), 120.3 (C-2), 74.5 (C-3), 72.5, 71.7 (C-4, C-6), 68.8 (C-5), 63.4 (C-7), 55.5 (OCH_3_). HR-ESI-MS positive mode (m/z): calc. for [M + H]^+^ = 701.2381, found: [M + H]^+^ = 701.2381; C_42_H_36_O_10_ (700.23).

#### 4.4.8. (*E*)-3,4,5,7-Tetra-*O*-Benzoyl-1-(3-Chlorophenyl)-1,2-Dideoxy-d-*gluco*-Hept-1-Enitol (**3f**) and (*E*)-3,4,6,7-Tetra-*O*-Benzoyl-1-(3-Chlorophenyl)-1,2-Dideoxy-d-*gluco*-Hept-1-Enitol (**4f**)

Isolated from a reaction of tosylhydrazone **1a** (0.10 g, 0.13 mmol), 3-chlorophenylboronic acid (1.5 equiv., 0.03 g, 0.19 mmol), and K_3_PO_4_ (3 equiv., 0.08 g, 0.39 mmol) according to General procedure I by column chromatography (1:4 EtOAc–hexane) to yield 60 mg white amorphous solid containing **3f** and **4f** in 2:1 ratio with two unidentified species. R_f_: 0.35 (1:2 EtOAc–hexane). 



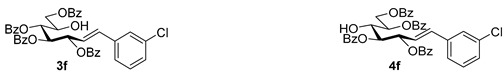



**3f**: ^1^H NMR (400 MHz, CDCl_3_) δ 8.18–7.82 (8H, m, aromatics), 7.64–7.02 (16H, m, aromatics), 6.70 (1H, d, *J*_1,2_ 15.9 Hz, H-1), 6.34 (1H, dd, *J*_2,3_ 6.9 Hz, H-2), 6.12–6.04 (2H, m, H-3, H-4), 5.76 (1H, dd, *J*_4,5_ 0.6, *J*_5,6_ 8.9 Hz, H-5), 4.54 (1H, dd, *J*_6,7a_ 3.0, *J*_7a,7b_ 11.9 Hz, H-7_a_), 4.34 (1H, dd, *J*_6,7b_ 5.7 Hz, H-7_b_), 4.23–4.14 (1H, m, H-6), 3.64 (1H, d, *J*_6,OH_ 3.9 Hz, OH). ^13^C NMR (90 MHz, CDCl_3_) δ 167.1, 166.7, 165.7, 165.5 (4 × CO), 134.9 (C-1), 138.4–123.4 (aromatics), 123.7 (C-2), 73.5 (C-3), 73.0 (C-4), 71.2 (C-5), 68.5 (C-6), 65.5 (C-7). HR-ESI-MS positive mode (m/z): calc. for [M + Na]^+^ = 727.1705, found: [M + Na]^+^ = 727.1706; C_41_H_33_ClO_9_ (704.18).

**4f**: ^1^H NMR (400 MHz, CDCl_3_) δ 8.18–7.79 (8H, m, aromatics), 7.69–7.06 (16H, m, aromatics), 6.91 (1H, d, *J*_1,2_ 15.8 Hz, H-1), 6.32 (1H, dd, *J*_2,3_ 8.0 Hz, H-2), 6.21 (1H, pseudo t, *J*_3,4_ 8.4 Hz, H-3), 5.81 (1H, dd, *J*_4,5_ 1.5 Hz, H-4), 5.43 (1H, ddd, *J*_6,7a_ 3.3, *J*_6,7b_ 4.3, *J*_5,6_ 8.0 Hz, H-6), 4.84 (1H, dd, *J*_7a,7b_ 12.4 Hz, H-7_a_), 4.73 (1H, dd, H-7_b_), 4.38–4.27 (1H, m, H-5), 3.34 (1H, d, *J*_5,OH_ 8.4 Hz, OH). ^13^C NMR (90 MHz, CDCl_3_) δ 167.1, 166.8, 165.8, 165.4 (4 × CO), 135.2 (C-1), 153.1–123.4 (aromatics), 124.4 (C-2), 74.4 (C-3), 72.4 (C-4), 71.7 (C-6), 68.4 (C-5), 63.4 (C-7). HR-ESI-MS positive mode (m/z): calc. for [M + Na]^+^ = 727.1705, found: [M + Na]^+^ = 727.1706; C_41_H_33_ClO_9_ (704.18).

#### 4.4.9. 3,4,5,7-Tetra-*O*-Benzoyl-1-(4-Chlorophenyl)-1,2-Dideoxy-d-*gluco*-Hept-1-Enitol (**3g**)

Prepared from tosylhydrazone **1a** (0.10 g, 0.13 mmol), 4-chlorophenylboronic acid (1.5 equiv., 0.03 g, 0.19 mmol), and K_3_PO_4_ (3 equiv., 0.08 g, 0.39 mmol) according to General procedure I. Purified by column chromatography (1:2 acetone–hexane) to yield 62 mg (68%) of **3g** as a white amorphous solid. R_f_: 0.36 (1:2 EtOAc–hexane); [α]_D_ + 9 (*c* 0.57, CH_2_Cl_2_). ^1^H NMR (400 MHz, CDCl_3_) δ 8.16–7.85 (8H, m, aromatics), 7.64–7.12 (16H, m, aromatics), 6.71 (1H, d, *J*_1,2_ 16.0 Hz, H-1), 6.34–6.24 (1H, m, H-2), 6.10–6.02 (2H, m, H-3, H-4), 5.74 (1H, dd, *J*_4,5_ 0.9, *J*_5,6_ 8.9 Hz, H-5), 4.53 (1H, dd, *J*_6,7a_ 2.9, *J*_7a,7b_ 11.9 Hz, H-7_a_), 4.34 (1H, dd, *J*_6,7b_ 5.7 Hz, H-7_b_), 4.21–4.10 (1H, m, H-6), 3.60 (1H, d, *J*_6,OH_ 5.1 Hz, OH). ^13^C NMR (100 MHz, CDCl_3_) δ 167.2, 166.8, 165.5, 165.4 (4 × CO), 135.2 (C-1), 134.7–127.2 (aromatics), 122.7 (C-2), 73.7 (C-3), 73.1 (C-4), 71.2 (C-5), 68.5 (C-6), 65.5 (C-7). HR-ESI-MS positive mode (m/z): calc. for [M + Na]^+^ = 727.1705, found: [M + Na]^+^ = 727.1703; C_41_H_33_ClO_9_ (704.18).



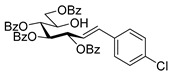



### 4.5. Characterization of Molecules ***7***–***9*** Isolated from the Reaction of Tosylhydrazone ***6*** and Phenylboronic Acid

#### 4.5.1. 3,4,5,7-Tetra-*O*-Acetyl-2,6-Anhydro-1-Deoxy-1-Phenyl-d-*glycero*-L-*manno*-Heptitol (**7**) 

Isolated from a reaction of tosylhydrazone **6** (0.10 g, 0.19 mmol), phenylboronic acid (20 equiv., 0.46 g, 3.78 mmol), and K_3_PO_4_ (10 equiv., 0.40 g, 1.89 mmol) according to General procedure I by column chromatography (1:4 EtOAc–hexane) to yield 6 mg (7%) of **7** as a white amorphous product. Optical rotation, NMR and MS spectra are identical with those reported [13].



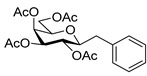



#### 4.5.2. (*E*)-3,5,6,7-Tetra-*O*-Acetyl-1,2-Dideoxy-1-Phenyl-d-*galacto*-Hept-1-Enitol (**8**)

Isolated from a reaction of tosylhydrazone **6** (0.30 g, 0.56 mmol), phenylboronic acid (20 equiv., 1.38 g, 11.35 mmol), and K_3_PO_4_ (10 equiv., 1.20 g, 5.68 mmol) according to General procedure I by column chromatography (1:4 EtOAc–hexane) to yield 12 mg (5%) of **8** as a white amorphous solid. R_f_: 0.15 (1:2 EtOAc–hexane); [α]_D_ + 40 (*c* 0.18, CHCl_3_). ^1^H NMR (400 MHz, CDCl_3_) δ 7.43–7.22 (5H, m, aromatics), 6.71 (1H, d, *J*_1,2_ 16.0 Hz, H-1), 6.32 (1H, dd, *J*_2,3_ 7.6 Hz, H-2), 5.54 (1H, ddd, *J*_6,7a_ 4.6, *J*_6,7b_ 7.7 Hz, H-6), 5.48 (1H, dd, *J*_3,4_ 1.1 Hz, H-3), 5.19 (1H, dd, *J*_5,6_ 1.7 Hz, H-5), 4.26 (1H, dd, *J*_7a,7b_ 11.8 Hz, H-7_a_), 4.05 (1H, dd, H-7_b_), 3.72 (1H, dd, *J*_4,5_ 9.6 Hz, H-4), 3.09 (1H, bs, OH), 2.16, 2.11, 2.10, 2.04 (12H, 4s, 4 × CH_3_). ^13^C NMR (90 MHz, CDCl_3_) δ 171.8, 170.6, 170.4, 170.0 (4 × CO), 134.9 (C-1), 136.7–123.2 (aromatics), 123.7 (C-2), 72.4 (C-3), 70.6 (C-4), 70.1, 70.0 (C-5, C-6), 62.8 (C-7), 21.3, 21.0, 20.8 (4 × CH_3_). HR-ESI-MS positive mode (m/z): calc. for [M + Na]^+^ = 445.1469, found: [M + Na]^+^ = 445.1470; C_21_H_26_O_9_ (422.16).



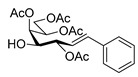



#### 4.5.3. (*E*)-3,4,6,7-Tetra-*O*-Acetyl-1,2-Dideoxy-1-Phenyl-d-*galacto*-Hept-1-Enitol (**9**) 

Prepared from tosylhydrazone **6** (0.30 g, 0.56 mmol), phenylboronic acid (20 equiv., 1.38 g, 11.35 mmol), and K_3_PO_4_ (10 equiv., 1.20 g, 5.68 mmol) according to General procedure I. Purified by column chromatography (1:4 EtOAc–hexane) to yield 180 mg (75%) of **9** as a white amorphous solid. R_f_: 0.10 (1:2 EtOAc–hexane); [α]_D_ + 37 (*c* 0.40, CHCl_3_). ^1^H NMR (400 MHz, CDCl_3_) δ 7.57–7.09 (5H, m, aromatics), 6.63 (1H, d, *J*_1,2_ 15.8 Hz, H-1), 6.03 (1H, dd, *J*_2,3_ 6.0 Hz, H-2), 5.88 (1H, dd, *J*_3,4_ 1.9 Hz, H-3), 5.22 (1H, ddd, *J*_6,7a_ 4.7, *J*_6,7b_ 7.8 Hz, H-6), 5.17 (1H, dd, *J*_4,5_ 9.7 Hz, H-4), 4.43 (1H, dd, *J*_7a,7b_ 11.7 Hz, H-7_a_), 4.17 (1H, dd, H-7_b_), 3.85 (1H, dd, *J*_5,6_ 1.5 Hz, H-5), 3.53 (1H, bs, OH), 2.21, 2.07, 2.04, 2.01 (12H, 4s, 4 × CH_3_). ^13^C NMR (100 MHz, CDCl_3_) δ 171.5, 171.2, 170.7, 170.0 (4 × CO), 133.0 (C-1), 136.0–121.2 (aromatics), 123.4 (C-2), 72.5 (C-3), 71.6 (C-4), 69.0, (C-6), 67.9 (C-5), 63.4 (C-7), 21.1, 20.8, 20.7 (4 × CH_3_). HR-ESI-MS positive mode (m/z): calc. for [M + Na]^+^ = 445.1469, found: [M + Na]^+^ = 445.1467; C_21_H_26_O_9_ (422.16).



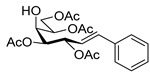



### 4.6. General Procedure II for the Synthesis of 1-Aryl-3,4,5,6,7-Penta-O-Benzoyl-1,2-Dideoxy-d-gluco-Hept-1-Enitols ***11*** and ***12***

A mixture of 1-aryl-tetra-*O*-benzoyl-1,2-dideoxy-d-*gluco*-hept-1-enitol (**3** and **4**, 1 mmol) and dry pyridine (6.3 mmol) were dissolved in dry chloroform (3 mL). Then, benzoyl–chloride (7 mmol) was added dropwise to the solution. The reaction mixture was stirred and heated at 80 °C. When TLC (1:2 EtOAc–hexane) showed complete consumption of the starting compound (~2 h), the mixture was cooled down. The organic layer was washed with 2M aqueous hydrogen chloride solution (1 × 3 mL), cold, saturated sodium hydrogen carbonate solution (1 × 3 mL), water (1 × 3 mL), and then dried on anhydrous magnesium sulfate. The solvent was removed under reduced pressure, and the residue was purified by column chromatography (1:2 EtOAc–hexane) to give hept-1-enitols.

#### 4.6.1. (*E*)-3,4,5,6,7-Penta-*O*-Benzoyl-1,2-Dideoxy-1-Phenyl-d-*gluco*-Hept-1-Enitol (**11a**)

Prepared from (*E*)-3,4,5,7-tetra-*O*-benzoyl-1,2-dideoxy-1-phenyl-d-*gluco*-hept-1-enitol **3a** and (*E*)-3,4,6,7-tetra-*O*-benzoyl-1,2-dideoxy-1-phenyl-d-*gluco*-hept-1-enitol **4a** (0.10 g, 0.15 mmol) according to General procedure II. Purified by column chromatography (1:2 EtOAc–hexane) to yield 104 mg (90%) of **11a** as a white amorphous solid. R_f_: 0.41 (1:2 EtOAc–hexane); [α]_D_ − 2 (*c* 0.50, CH_2_Cl_2_). ^1^H NMR (400 MHz, CDCl_3_) δ 8.24–7.84 (8H, m, aromatics), 7.66–7.17 (17H, m, aromatics), 6.80 (1H, d, *J*_1,2_ 15.9 Hz, H-1), 6.40–6.29 (1H, m, H-2), 6.18 (1H, dd, *J*_4,5_ 2.0 Hz, H-5), 6.12–6.04 (2H, m, H-3, H-4), 5.91 (1H, ddd, *J*_6,7a_ 3.6, *J*_6,7b_ 5.9, *J*_5,6_ 7.2 Hz, H-6), 4.82 (1H, dd, *J*_7a,7b_ 12.3 Hz, H-7_a_), 4.55 (1H, dd, H-7_b_). ^13^C NMR (100 MHz, CDCl_3_) δ 166.1, 165.7, 165.5, 165.4, 165.3 (5 × CO), 136.7 (C-1), 135.8–127.0 (aromatics), 122.0 (C-2), 73.8 (C-3), 71.8 (C-4), 69.9, 69.7 (C-5, C-6), 62.8 (C-7). HR-ESI-MS positive mode (m/z): calc. for [M + Na]^+^ = 797.2357, found: [M + Na]^+^ = 797.2355; C_48_H_38_O_10_ (774.25).



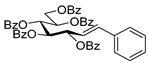



#### 4.6.2. (*E*)-3,4,5,6,7-Penta-*O*-Benzoyl-1-(4-Dibenzo[b,d]furanyl)-1,2-Dideoxy-d-*gluco*-Hept-1-Enitol (**11b**) 

Prepared from (*E*)-3,4,5,7-tetra-*O*-benzoyl-1-(4-dibenzo[*b,d*]furanyl)-1,2-dideoxy-d-*gluco*-hept-1-enitol **3c** and (*E*)-3,4,6,7-tetra-*O*-benzoyl-1-(4-dibenzo[*b,d*]furanyl)-1,2-dideoxy-d-*gluco*-hept-1-enitol **4c** (0.05 g, 0.06 mmol), according to General procedure II. Purified by column chromatography (1:2 EtOAc–hexane) to yield 28 mg (55%) of **11b** as a yellow amorphous solid. R_f_: 0.39 (1:2 EtOAc–hexane); [α]_D_ − 1 (*c* 0.48, CH_2_Cl_2_). ^1^H NMR (400 MHz, CDCl_3_) δ 8.44–6.81 (32H, m, aromatics), 7.08–7.04 (2H, m, H-1, H-2), 6.28 (1H, dd, *J*_4,5_ 2.0 Hz, H-5), 6.20–6.14 (2H, m, H-3, H-4), 5.92 (1H, ddd, *J*_6,7a_ 3.8, *J*_6,7b_ 5.8, *J*_5,6_ 7.1 Hz, H-6), 4.82 (1H, dd, *J*_7a,7b_ 12.2 Hz, H-7_a_), 4.53 (1H, dd, H-7_b_). ^13^C NMR (100 MHz, CDCl_3_) δ 166.1, 165.7, 165.5, 165.4, 165.3 (5 × CO), 130.9 (C-1), 162.8–110.9 (aromatics), 125.8 (C-2), 74.0 (C-3), 71.8 (C-4), 69.9 (C-6), 69.6 (C-5), 62.8 (C-7). HR-ESI-MS positive mode (m/z): calc. for [M + Na]^+^ = 887.2463, found: [M + Na]^+^ = 887.2460; C_54_H_40_O_11_ (864.26).



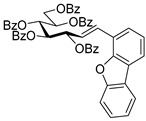



#### 4.6.3. (*E*)-3,4,5,6,7-Penta-*O*-Acetyl-1,2-Dideoxy-1-Phenyl-d-*galacto*-Hept-1-Enitol (**12**) 

3,4,6,7-Tetra-*O*-acetyl-1,2-dideoxy-1-phenyl-d-*galacto*-hept-1-enitol (**9**, 0.12 g, 0.29 mmol) was dissolved in dry pyridine (1 mL) and cooled to 0 °C. Then, acetic anhydride (1.5 equiv., 0.04 mL, 0.04 g, 0.43 mmol) was added dropwise to the solution. The reaction mixture was stirred for a day at room temperature and the pyridine was evaporated. The residue was dissolved in dichloromethane and washed with water (1 × 2 mL), then dried on anhydrous magnesium sulfate. The solution was concentrated under reduced pressure and traces of pyridine were removed by repeated co-evaporations with toluene to yield 122 mg (91%) of **12** as a white amorphous solid. R_f_: 0.36 (1:2 EtOAc–hexane); [α]_D_ + 115 (*c* 0.02, CH_2_Cl_2_). ^1^H NMR (400 MHz, CDCl_3_) δ 7.57–7.06 (5H, m, aromatics), 6.58 (1H, dd, *J*_Ar,1_ 0.7, *J*_1,2_ 15.9 Hz, H-1), 5.97 (1H, dd, *J*_2,3_ 6.1 Hz, H-2), 5.67–5.59 (1H, m, H-3), 5.45 (1H, dd, *J*_5,6_ 1.8 Hz, H-5), 5.41–5.31 (1H, m, H-6), 5.37 (1H, dd, *J*_3,4_ 2.5, *J*_4,5_ 10.0 Hz, H-4), 4.29 (1H, dd, *J*_6,7a_ 5.0, *J*_7a,7b_ 11.6 Hz, H-7_a_), 3.88 (1H, dd, *J*_6,7a_ 7.5 Hz, H-7_b_), 2.14, 2.10, 2.08, 2.04, 202 (15H, 5s, 5 × CH_3_). ^13^C NMR (90 MHz, CDCl_3_) δ 170.5, 170.3, 170.1, 169.8 (5 × CO), 133.5 (C-1), 136.5–122.2 (aromatics), 122.9 (C-2), 71.1 (C-3), 69.5 (C-4), 68.1 (C-5), 68.0 (C-6), 62.3 (C-7), 21.0, 20.8, 20.7 (5 × CH_3_). HR-ESI-MS positive mode (m/z): calc. for [M + Na]^+^ = 487.1575, found: [M + Na]^+^ = 487.11577; C_23_H_28_O_10_ (464.17).



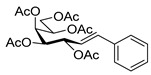



### 4.7. General Procedure III for the Synthesis of Anhydro-Aldose Tosylhydrazones (C-(2,3,4,6-Tetra-O-Alkyl-β-d-Glycopyranosyl) Formaldehyde Tosylhydrazones) (***17**, **24***)

Raney-nickel (1.5 g, an aqueous suspension, Merck) was added at room temperature to a vigorously stirred solution of pyridine (6 mL), acetic acid (4 mL), and water (4 mL). Then, sodium hypophosphite (0.75 g, 8.50 mmol), tosylhydrazine (0.37 g, 2.00 mmol), and nitrile (**16β** [52] or **23**) (1.00 mmol) were added to the mixture. When TLC (2:1 EtOAc–hexane) indicated complete consumption of the starting compound, the insoluble material was filtered off through a pad of celite and washed with dichloromethane (10 mL). The organic layer of the filtrate was separated, washed with water (3 mL), 10% aqueous hydrogen chloride solution (2 × 3 mL), cold, saturated sodium hydrogen carbonate solution (2 × 3 mL), water (3 mL), and then dried on anhydrous magnesium sulfate. The solution was concentrated under reduced pressure, and traces of pyridine were removed by repeated co-evaporations with toluene. The residue was purified by silica gel column chromatography with eluents indicated for the particular compounds to give anhydro-aldose tosylhydrazones **17** or **24**.

#### 2,6-Anhydro-3,4,5,7-Tetra-*O*-Methyl-d-*glycero*-d-*gulo*-Heptose Tosylhydrazone (C-(2,3,4,6-Tetra-*O*-Methyl-β-d-Glucopyranosyl) Formaldehyde Tosylhydrazone) (**17**) 

Prepared from cyanide **16β** [52] (1.00 g, 4.08 mmol) according to General procedure III. Purified by column chromatography (1:2 EtOAc–hexane) to yield 1.02 g (60%) two unidentified isomers **17-1** and **17-2** in 1:3 ratio as a colourless oil. 



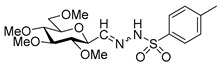



**17-1** R_f_: 0.11 (1:1 EtOAc–hexane). ^1^H NMR (360 MHz, CDCl_3_) δ 7.92 (1H, bs, NH), 7.86–7.76 (2H, m, aromatics), 7.31 (2H, d, *J* 8.2 Hz, aromatics), 7.05 (1H, d, *J*_1,2_ 6.0 Hz, H-1), 3.74 (1H, dd, *J*_2,3_ 9.5 Hz, H-2), 3.61–3.45 and 3.32–3.00 (6H, m, H-3–H-7_b_), 3.63, 3.52, 3.35, 3.25 (12H, 4s, 4 × CH_3_), 2.41 (3H, s, CH_3_-Ts). ^13^C NMR (90 MHz, CDCl_3_) δ 146.7 (C-1), 144.3–127.5 (aromatics), 88.5, 82.7, 79.4, 78.6, 74.1 (C-2–C-6), 71.0 (C-7), 60.9, 60.7, 60.6, 59.3 (4 × CH_3_), 21.6 (CH_3_-Ts). HR-ESI-MS positive mode (m/z): calcd. for [M + H]^+^ = 417.1692, found: [M + H]^+^ = 417.1694; C_18_H_28_N_2_O_7_S (416.16).

**17-2** R_f_: 0.10 (1:1 EtOAc–hexane). ^1^H NMR (360 MHz, CDCl_3_) δ 9.31 (1H, bs, NH), 7.83 (2H, d, *J* 8.2 Hz, aromatics), 7.31 (2H, d, *J* 8.2 Hz, aromatics), 6.80 (1H, d, *J*_1,2_ 4.6 Hz, H-1), 4.02 (1H, dd, *J*_2,3_ 10.3 Hz, H-2), 3.61–3.45 and 3.32–3.00 (6H, m, H-3–H-7_b_), 3.63, 3.52, 3.41, 3.30 (4s, 12H, 4 × CH_3_), 2.41 (s, 3H, CH_3_-Ts). ^13^C NMR (90 MHz, CDCl_3_) δ 146.2 (C-1), 144.3–127.5 (aromatics), 87.9, 81.2, 79.4, 78.4, 77.9 (C-2–C-6), 71.3 (C-7), 60.8, 60.4, 59.8, 59.2 (4 × CH_3_), 21.6 (CH_3_-Ts). HR-ESI-MS positive mode (m/z): calcd. for [M + H]^+^ = 417.1692, found: [M + H]^+^ = 417.1694; C_18_H_28_N_2_O_7_S (416.16).

### 4.8. Characterization of Anhydro-Heptitols ***18***

#### 4.8.1. 2,6-Anhydro-1-Deoxy-3,4,5,7-Tetra-*O*-Methyl-1-Phenyl-d-*glycero*-d-*gulo*-Heptitol (**18a**) 

Isolated from a reaction of tosylhydrazone **17** (0.05 g, 0.13 mmol), phenylboronic acid (1.5 equiv., 0.02 g, 0.19 mmol), and K_3_PO_4_ (3 equiv., 0.08 g, 0.39 mmol) according to General procedure I by column chromatography (1:2 EtOAc–hexane) to yield 18 mg amorphous solid containing **18a** and **21** in 1.3:1 ratio. R_f_: 0.50 (1:2 EtOAc–hexane). ^1^H NMR (400 MHz, CDCl_3_) δ 8.34–6.74 (5H, m, aromatics), 3.65 (3H, s, *CH_3_*OC-4), 3.59 (3H, s, *CH_3_*OC-3), 3.53 (3H, s, *CH_3_*OC-5), 3.55–3.50 (2H, m, H-7_a_, H-7_b_), 3.36 (3H, s, *CH_3_*OC-7), 3.30 (1H, ddd, *J*_1a,2_ 2.4, *J*_1b,2_ 8.8, *J*_2,3_ 8.9 Hz, H-2), 3.23–3.15 (2H, m, H-4, H-5), 3.12 (1H, ddd, *J*_6,7a_ 2.0, *J*_6,7b_ 4.0, *J*_5,6_ 9.8 Hz, H-6), 3.07 (1H, dd, *J*_1a,1b_ 14.3 Hz, H-1_a_), 2.90 (1H, pseudo t, *J*_3,4_ 9.0 Hz, strongly coupled, H-3), 2.74 (1H, dd, H-1_b_). ^13^C NMR (90 MHz, CDCl_3_) δ 139.4–126.1 (aromatics), 89.2 (C-4), 83.7 (C-3), 80.3 (C-2), 80.1 (C-5), 78.8 (C-6), 71.5 (C-7), 60.8 (*CH_3_*OC-4) 60.7 (*CH_3_*OC-3), 60.4 (*CH_3_*OC-5), 59.5 (*CH_3_*OC-7), 37.9 (C-1). HR-ESI-MS positive mode (m/z): calc. for [M + Na]^+^ = 333.1672, found: [M + Na]^+^ = 333.1672; C_17_H_26_O_5_ (310.39).



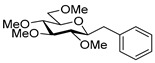



#### 4.8.2. 2,6-Anhydro-1-Deoxy-1-(4-Dibenzo[b,d]furanyl)-3,4,5,7-Tetra-*O*-Methyl-d-*glycero*-d-*gulo*-Heptitol (**18b**) 

Isolated from a reaction of tosylhydrazone **17** (0.05 g, 0.13 mmol), dibenzo[*b,d*]furan-4-ylboronic acid (1.5 equiv., 0.04 g, 0.19 mmol), and K_3_PO_4_ (3 equiv., 0.08 g, 0.39 mmol) according to General procedure I by column chromatography (1:4 EtOAc–hexane) to yield 4 mg (8%) of **18b** as a white amorphous solid. R_f_: 0.47 (1:2 EtOAc–hexane). ^1^H NMR (500 MHz, CDCl_3_) δ 7.93 (1H, d, *J* 7.7 Hz, aromatic), 7.80 (1H, dd, *J* 1.1, 7.7 Hz, aromatic), 7.58 (1H, d, *J* 8.2 Hz, aromatic), 7.47–7.39 (2H, m, aromatics), 7.35–7.30 (1H, m, aromatic), 7.29–7.23 (1H, m, aromatics), 3.67 (3H, s, *CH_3_*OC-4), 3.62 (3H, s, *CH_3_*OC-3), 3.58 (1H, ddd, *J*_1a,2_ 2.9, *J*_1b,2_ 8.9, *J*_2,3_ 9.2 Hz, H-2), 3.54 (1H, dd, H-1_a_), 3.53 (3H, s, *CH_3_*OC-5), 3.48 (1H, dd, H-7_a_), 3.46 (1H, dd, *J*_7a,7b_ 11.2 Hz, H-7_b_), 3.28 (3H, s, *CH_3_*OC-7), 3.26 (1H, pseudo t, *J*_3,4_ 8.7 Hz, H-4), 3.21 (1H, pseudo t, *J*_4,5_ 8.8 Hz, H-5), 3.14 (1H, ddd, *J*_6,7a_ 2.5, *J*_6,7b_ 3.4, *J*_5,6_ 9.5 Hz, H-6), 3.09 (1H, dd, *J*_1a,1b_ 14.4 Hz, H-1_b_) 3.02 (1H, pseudo t, *J*_3,4_ 8.9 Hz, H-3). ^13^C NMR (90 MHz, CDCl_3_) δ 129.4–110.9 (aromatics), 89.2 (C-4), 84.2 (C-3), 80.2 (C-5), 79.0, 78.9 (C-2, C-6), 71.5 (C-7), 60.9 (*CH_3_*OC-4), 60.8 (*CH_3_*OC-3), 60.5 (*CH_3_*OC-5), 59.5 (*CH_3_*OC-7), 32.0 (C-1). HR-ESI-MS positive mode (m/z): calc. for [M + Na]^+^ = 423.1778, found: [M + Na]^+^ = 423.1777; C_23_H_28_O_6_ (400.19).



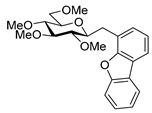



#### 4.8.3. 2,6-Anhydro-1-Deoxy-3,4,5,7-Tetra-*O*-Methyl-1-(4-Trifluoromethylphenyl)-d-*glycero*-d-*gulo*-Heptitol (**18c**) 

Prepared from tosylhydrazone **17** (0.05 g, 0.13 mmol), (4-trifluoromethly)phenylboronic acid (1.5 equiv., 0.04 g, 0.39 mmol), and K_3_PO_4_ (3 equiv., 0.08 g, 0.39 mmol) according to General procedure I. Purified by column chromatography (1:3 EtOAc–hexane) to yield 22 mg (45%) of **18c** as a white amorphous solid. R_f_: 0.50 (1:2 EtOAc–hexane); [α]_D_ − 6 (*c* 0.30, CH_2_Cl_2_). ^1^H NMR (400 MHz, CDCl_3_) δ 7.52 (2H, d, *J* 8.1 Hz, aromatics), 7.38 (2H, d, *J* 8.1 Hz, aromatics), 3.65 (3H, s, *CH_3_*OC-4), 3.59 (3H, s, *CH_3_*OC-3), 3.54 (1H, dd, H-7_a_), 3.53 (3H, s, *CH_3_*OC-5), 3.50 (1H, dd, *J*_7a,7b_ 11.1 Hz, H-7_b_), 3.36 (3H, s, *CH_3_*OC-7), 3.29 (1H, ddd, *J*_1a,2_ 2.3, *J*_1b,2_ 8.9, *J*_2,3_ 9.2 Hz, H-2), 3.24–3.08 (3H, m, H-1_a_, H-4, H-5), 3.13 (1H, ddd, *J*_6,7a_ 2.0, *J*_6,7b_ 3.9, *J*_5,6_ 9.8 Hz, H-6), 2.88 (1H, pseudo t, *J*_3,4_ 8.9 Hz, strongly coupled, H-3), 2.80 (1H, dd, *J*_1a,1b_ 14.2 Hz, H-1_b_). ^13^C NMR (100 MHz, CDCl_3_) δ 143.7–124.1 (aromatics), 89.2 (C-4), 83.6 (C-3), 80.0 (C-2), 79.8 (C-5), 78.8 (C-6), 71.4 (C-7), 60.8 (*CH_3_*OC-4), 60.8 (*CH_3_*OC-3), 60.5 (*CH_3_*OC-5), 59.5 (*CH_3_*OC-7), 37.7 (C-1). HR-ESI-MS positive mode (m/z): calc. for [M + H]^+^ = 379.1727, found: [M + H]^+^ = 379.1727; C_18_H_25_F_3_O_5_ (378.17).



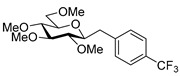



#### 4.8.4. 2,6-Anhydro-1-Deoxy-1-(4-Fluorophenyl)-3,4,5,7-Tetra-*O*-Methyl-d-*glycero*-d-*gulo*-Heptitol (**18d**) 

Isolated from a reaction of tosylhydrazone **17** (0.05 g, 0.13 mmol), 4-fluorophenylboronic acid (1.5 equiv., 0.03 g, 0.19 mmol), and K_3_PO_4_ (3 equiv., 0.08 g, 0.39 mmol) according to General procedure I by column chromatography (1:4 EtOAc–hexane) to yield 6 mg (14%) of **18d** as a white amorphous solid. R_f_: 0.41 (1:2 EtOAc–hexane); [α]_D_ + 0.5 (*c* 0.20, CH_2_Cl_2_). ^1^H NMR (400 MHz, CDCl_3_) δ 7.22 (2H, dd, *J* 5.6, 8.6 Hz, aromatics), 6.94 (2H, t, *J* 8.8 Hz, aromatics), 3.65 (3H, s, *CH_3_*OC-4), 3.58 (3H, s, *CH_3_*OC-3), 3.54 (1H, dd, H-7_a_), 3.53 (3H, s, *CH_3_*OC-5), 3.50 (1H, dd, *J*_7a,7b_ 10.8 Hz, H-7_b_), 3.37 (3H, s, *CH_3_*OC-7), 3.24 (1H, ddd, *J*_1a,2_ 2.1, *J*_1b,2_ 8.8, *J*_2,3_ 9.1 Hz, H-2), 3.21–3.13 (2H, m, H-4, H-5), 3.12 (1H, ddd, *J*_6,7a_ 1.9, *J*_6,7b_ 3.6, *J*_5,6_ 8.7 Hz, H-6), 3.04 (1H, dd, *J*_1a,1b_ 14.3 Hz, H-1_a_), 2.87 (1H, pseudo t, *J*_3,4_ 9.0 Hz, strongly coupled, H-3), 2.71 (1H, dd, H-1_b_). ^13^C NMR (90 MHz, CDCl_3_) δ 131.4–109.0 (aromatics), 89.3 (C-4), 83.6 (C-3), 80.2, (C-2), 80.1 (C-5), 78.8 (C-6), 71.5 (C-7), 60.8 (*CH_3_*OC-3, *CH_3_*OC-4), 60.5 (*CH_3_*OC-5), 59.5 (*CH_3_*OC-7), 37.1 (C-1). HR-ESI-MS positive mode (m/z): calc. for [M + H]^+^ = 329.1759, found: [M + Na]^+^ = 329.1759; C_17_H_25_FO_5_ (328.17).



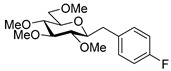



#### 4.8.5. 2,6-Anhydro-1-(3-Chlorophenyl)-1-Deoxy-3,4,5,7-Tetra-*O*-Methyl-d-*glycero*-d-*gulo*-Heptitol (**18e**) 

Prepared from tosylhydrazone **17** (0.05 g, 0.13 mmol), 3-chorophenylboronic acid (1.5 equiv., 0.03 g, 0.19 mmol), and K_3_PO_4_ (3 equiv., 0.08 g, 0.39 mmol) according to General procedure I. Purified by column chromatography (1:2 EtOAc–hexane) to yield 13 mg (29%) of **18e** as a pale-yellow amorphous solid. R_f_: 0.48 (1:2 EtOAc–hexane); [α]_D_ − 3 (*c* 0.24, CH_2_Cl_2_). ^1^H NMR (400 MHz, CDCl_3_) δ 7.32–7.24 (1H, m, aromatic), 7.23–7.08 (3H, m, aromatics), 3.65 (3H, s, *CH_3_*OC-4), 3.59 (3H, s, *CH_3_*OC-3), 3.56 (1H, dd, *J*_7a,7b_ 11.0 Hz, H-7_a_), 3.53 (3H, s, *CH_3_*OC-5), 3.51 (1H, dd, H-7_b_), 3.37 (3H, s, *CH_3_*OC-7), 3.27 (1H, ddd, *J*_1a,2_ 2.3, *J*_1b,2_ 8.8, *J*_2,3_ 9.1 Hz, H-2), 3.23–3.14 (2H, m, H-4, H-5), 3.13 (1H, ddd, *J*_6,7a_ 1.6, *J*_6,7b_ 3.4, *J*_5,6_ 8.6 Hz, H-6), 3.04 (1H, dd, *J*_1a,1b_ 14.3 Hz, H-1_a_), 2.87 (1H, pseudo t, *J*_3,4_ 8.8 Hz, H-3), 2.71 (1H, dd, H-1_b_). ^13^C NMR (90 MHz, CDCl_3_) δ 141.3–126.0 (aromatics), 89.2 (C-4), 83.5 (C-3), 80.1 (C-2), 79.9 (C-5), 78.8 (C-6), 71.5 (C-7), 60.9 (*CH_3_*OC-4), 60.8 (*CH_3_*OC-3), 60.5 (*CH_3_*OC-5), 59.6 (*CH_3_*OC-7), 37.6 (C-1). HR-ESI-MS positive mode (m/z): calc. for [M + H]^+^ = 345.1463, found: [M + H]^+^ = 345.1460; C_17_H_25_ClO_5_ (344.14).



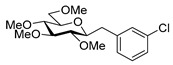



#### 4.8.6. 2,6-Anhydro-1-(4-Bromophenyl)-1-Deoxy-3,4,5,7-Tetra-*O*-Methyl-d-*glycero*-d-*gulo*-Heptitol (**18f**)

Prepared from tosylhydrazone **17** (0.05 g, 0.13 mmol), 4-bromophenylboronic acid (1.5 equiv., 0.04 g, 0.19 mmol), and K_3_PO_4_ (3 equiv., 0.08 g, 0.39 mmol) according to General procedure I. Purified by column chromatography (1:4 EtOAc–hexane) to yield 11 mg (22%) of **18f** as a white amorphous solid. R_f_: 0.53 (1:2 EtOAc–hexane); [α]_D_ − 6 (*c* 0.21, CH_2_Cl_2_). ^1^H NMR (400 MHz, CDCl_3_) δ 7.38 (2H, d, *J* 8.4 Hz, aromatics), 7.14 (2H, d, *J* 8.4 Hz, aromatics), 3.65 (3H, s, *CH_3_*OC-4), 3.58 (3H, s, *CH_3_*OC-3), 3.53 (3H, s, *CH_3_*OC-5), 3.53 (1H, dd, H-7_a_), 3.49 (1H, dd, *J*_7a,7b_ 10.8 Hz, H-7_b_), 3.37 (3H, s, *CH_3_*OC-7), 3.24 (1H, ddd, *J*_1a,2_ 2.2, *J*_1b,2_ 8.9, *J*_2,3_ 9.1 Hz, H-2), 3.21–3.13 (2H, m, H-4, H-5), 3.11 (1H, ddd, *J*_6,7a_ 2.1, *J*_6,7b_ 3.5, *J*_5,6_ 9.8 Hz, H-6), 3.02 (1H, dd, *J*_1a,1b_ 14.3 Hz, H-1_a_), 2.87 (1H, pseudo t, *J*_3,4_ 9.0 Hz, strongly coupled, H-3), 2.69 (1H, dd, H-1_b_). ^13^C NMR (100 MHz, CDCl_3_) δ 138.4–119.3 (aromatics), 89.2 (C-4), 83.5 (C-3), 80.0 (C-2, C-5), 78.8 (C-6), 71.4 (C-7), 60.9 (*CH_3_*OC-4), 60.8 (*CH_3_*OC-3), 60.5 (*CH_3_*OC-5), 59.5 (*CH_3_*OC-7), 37.3 (C-1). HR-ESI-MS positive mode (m/z): calc. for [M + H]^+^ = 389.0958, found: [M + H]^+^ = 389.0959; C_17_H_25_BrO_5_ (389.29).



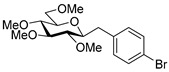



#### 4.8.7. 2,6-Anhydro-1-Deoxy-3,4,5,7-Tetra-*O*-Methyl-1-(4-Nitrophenyl)-d-*glycero*-d-*gulo*-Heptitol (**18g**) 

Prepared from tosylhydrazone **17** (0.05 g, 0.13 mmol), 4-nitrophenylboronic acid (1.5 equiv., 0.03 g, 0.19 mmol), and K_3_PO_4_ (3 equiv., 0.08 g, 0.39 mmol) according to General procedure I. Purified by column chromatography (1:2 EtOAc–hexane) to yield 21 mg (46%) of **18g** as a yellow amorphous solid. R_f_: 0.26 (1:2 EtOAc–hexane); [α]_D_ + 3 (*c* 0.22, CH_2_Cl_2_). ^1^H NMR (400 MHz, CDCl_3_) δ 8.13 (2H, d, *J* 8.7 Hz, aromatics), 7.43 (2H, d, *J* 8.7 Hz, aromatics), 3.66 (3H, s, *CH_3_*OC-4), 3.60 (3H, s, *CH_3_*OC-3), 3.53 (3H, s, *CH_3_*OC-5), 3.53 (1H, dd, H-7_a_), 3.49 (1H, dd, *J*_7a,7b_ 10.8 Hz, H-7_b_), 3.37 (3H, s, *CH_3_*OC-7), 3.29 (1H, ddd, *J*_1a,2_ 2.4, *J*_1b,2_ 9.0, *J*_2,3_ 9.2 Hz, H-2), 3.23–3.14 (3H, m, H-1_a_, H-4, H-5), 3.12 (1H, ddd, *J*_6,7a_ 1.3, *J*_6,7b_ 3.1, *J*_5,6_ 9.6 Hz, H-6), 2.89 (1H, pseudo t, *J*_3,4_ 8.8 Hz, strongly coupled, H-3), 2.85 (1H, dd, H-1_b_). ^13^C NMR (90 MHz, CDCl_3_) δ 147.6–121.1 (aromatics), 89.2 (C-4), 83.5 (C-3), 80.0 (C-2), 79.6 (C-5), 78.8 (C-6), 71.4 (C-7), 60.9 (*CH_3_*OC-3, *CH_3_*OC-4), 60.5 (*CH_3_*OC-5), 59.5 (*CH_3_*OC-7), 37.8 (C-1). HR-ESI-MS positive mode (m/z): calc. for [M + H]^+^ = 356.1704, found: [M + H]^+^ = 356.1704; C_17_H_25_NO_7_ (355.16).



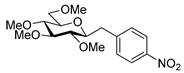



#### 4.8.8. 2,6-Anhydro-1-Deoxy-3,4,5,7-Tetra-*O*-Methyl-1-(4-Methoxyphenyl)-d-*glycero*-d-*gulo*-Heptitol (**18h**)

Isolated from a reaction of tosylhydrazone **17** (0.05 g, 0.13 mmol), 4-methoxyphenylboronic acid (1.5 equiv., 0.03 g, 0.19 mmol), and K_3_PO_4_ (3 equiv., 0.08 g, 0.39 mmol) according to General procedure I by column chromatography (1:2 EtOAc–hexane) to yield 4 mg (9%) of **18h** as a pale-yellow amorphous solid. R_f_: 0.41 (1:2 EtOAc–hexane); [α]_D_ + 5 (*c* 0.57, CH_2_Cl_2_). ^1^H NMR (400 MHz, CDCl_3_) δ 7.19 (2H, d, *J* 8.6 Hz, aromatics), 6.81 (2H, d, *J* 8.7 Hz, aromatics), 3.79 (3H, s, OCH_3_), 3.65 (3H, s, *CH_3_*OC-4), 3.59 (3H, s, *CH_3_*OC-3), 3.55 (1H, dd, *J*_7a,7b_ 10.9 Hz, H-7_a_), 3.53 (3H, s, *CH_3_*OC-5), 3.50 (1H, dd, H-7_b_), 3.38 (3H, s, *CH_3_*OC-7), 3.24 (1H, ddd, *J*_1a,2_ 2.3, *J*_1b,2_ 8.8, *J*_2,3_ 9.1 Hz, H-2), 3.21–3.14 (2H, m, H-4, H-5), 3.12 (1H, ddd, *J*_6,7a_ 2.0, *J*_6,7b_ 3.9, *J*_5,6_ 9.8 Hz, H-6), 3.01 (1H, dd, *J*_1a,1b_ 14.3 Hz, H-1_a_), 2.88 (1H, pseudo t, *J*_3,4_ 9.0 Hz, strongly coupled, H-3), 2.69 (1H, dd, H-1_b_). ^13^C NMR (90 MHz, CDCl_3_) δ 162.4–109.6 (aromatics), 89.3 (C-4), 83.7 (C-3), 80.5 (C-2), 80.1 (C-5), 78.8 (C-6), 71.6 (C-7), 60.8 (*CH_3_*OC-3, *CH_3_*OC-4), 60.5 (*CH_3_*OC-5), 59.6 (*CH_3_*OC-7), 55.4 (OCH_3_), 37.0 (C-1). HR-ESI-MS positive mode (m/z): calc. for [M + H]^+^ = 341.1959, found: [M + H]^+^ = 341.1957; C_18_H_28_O_5_ (340.42).



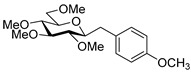



#### 4.8.9. 2,6-Anhydro-1-Deoxy-3,4,5,7-Tetra-*O*-Methyl-1-(4-Methylphenyl)-d-*glycero*-d-*gulo*-Heptitol (**18i**)

Isolated from a reaction of tosylhydrazone **17** (0.05 g, 0.13 mmol), 4-methylphenylboronic acid (1.5 equiv., 0.03 g, 0.19 mmol), and K_3_PO_4_ (3 equiv., 0.08 g, 0.39 mmol) according to General procedure I by column chromatography (1:2 EtOAc–hexane) to yield 11 mg white amorphous solid containing **18i** and **21** in 3:1 ratio. R_f_: 0.48 (1:2 EtOAc–hexane); [α]_D_ + 0.5 (*c* 0.08, CH_2_Cl_2_). ^1^H NMR (400 MHz, CDCl_3_) δ 7.16 (2H, d, *J* 7.9 Hz, aromatics), 7.07 (2H, d, *J* 7.9 Hz, aromatics), 3.65 (3H, s, *CH_3_*OC-4), 3.59 (3H, s, *CH_3_*OC-3), 3.53 (3H, s, *CH_3_*OC-5), 3.55–3.50 (2H, m, H-7_a_, H-7_b_), 3.37 (3H, s, *CH_3_*OC-7), 3.26 (1H, ddd, *J*_1a,2_ 2.1, *J*_1b,2_ 8.8, *J*_2,3_ 9.1 Hz, H-2), 3.23–3.14 (2H, m, H-4, H-5), 3.11 (1H, ddd, *J*_6,7a_ 1.9, *J*_6,7b_ 3.6, *J*_5,6_ 9.7 Hz, H-6), 3.03 (1H, dd, *J*_1a,1b_ 14.3 Hz, H-1_a_), 2.88 (1H, pseudo t, *J*_3,4_ 9.0 Hz, strongly coupled, H-3), 2.70 (1H, dd, H-1_b_), 2.31 (3H, s, CH_3_). ^13^C NMR (100 MHz, CDCl_3_) δ 136.2–128.6 (aromatics), 89.3 (C-4), 83.7 (C-3), 80.4 (C-2), 80.1 (C-5), 78.8 (C-6), 71.5 (C-7), 60.8 (*CH_3_*OC-3, *CH_3_*OC-4), 60.5 (*CH_3_*OC-5), 59.6 (*CH_3_*OC-7), 37.5 (C-1), 21.2 (CH_3_). HR-ESI-MS positive mode (m/z): calc. for [M + H]^+^ = 325.2010, found: [M + H]^+^ = 325.2008; C_18_H_28_O_5_ (324.42).



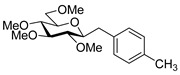



### 4.9. Characterization of Heptenitols ***19*** and ***20***

#### 4.9.1. (*E*)-1,2-Dideoxy-3,4,5,7-Tetra-*O*-Methyl-1-Phenyl-d-*gluco*-Hept-1-Enitol (**19a**) and (*Z*)-1,2-Dideoxy-3,4,5,7-Tetra-*O*-Methyl-1-Phenyl-d-*gluco*-Hept-1-Enitol (**20a**)

Isolated from a reaction of tosylhydrazone **17** (0.05 g, 0.13 mmol), phenylboronic acid (1.5 equiv., 0.02 g, 0.19 mmol), and K_3_PO_4_ (3 equiv., 0.08 g, 0.39 mmol) according to General procedure I by column chromatography (1:2 EtOAc–hexane) to yield 29 mg white amorphous solid containing **19a** and **20a** in 9:1 ratio. R_f_: 0.16 (1:2 EtOAc–hexane), [α]_D_ + 28 (*c* 0.16, CH_2_Cl_2_). 



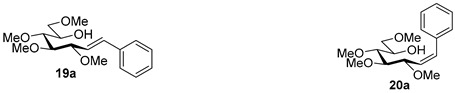



**19a**: ^1^H NMR (500 MHz, CDCl_3_) δ 7.42 (2H, d, *J* 7.6 Hz, aromatics), 7.38–7.30 (2H, m, aromatics), 7.29–7.23 (1H, m, aromatic), 6.63 (1H, d, *J*_1,2_ 16.0 Hz, H-1), 6.16 (1H, dd, *J*_2,3_ 8.2 Hz, H-2), 4.05 (1H, dd, *J*_3,4_ 6.0 Hz, H-3), 3.96 (1H, ddd, *J*_6,7a_ 3.9, *J*_6,7b_ 5.5, *J*_5,6_ 6.7 Hz, H-6), 3.60 (3H, s, *CH_3_*OC-4), 3.59–3.50 (3H, m, H-4, H-7_a_, H-7_b_), 3.40 (6H, 2s, *CH_3_*OC-5, *CH_3_*OC-7), 3.40–3.37 (1H, m, H-5), 3.37 (3H, s, *CH_3_*OC-3), 3.32 (1H, bs, OH). ^13^C NMR (125 MHz, CDCl_3_) δ 134.0 (C-1), 137.0–126.3 (aromatics), 126.7 (C-2), 83.8 (C-4), 83.4 (C-3), 79.8 (C-5), 73.8 (C-7), 70.2 (C-6), 60.8 (*CH_3_*OC-4), 59.4 (*CH_3_*OC-5), 59.2 (*CH_3_*OC-7), 56.8 (*CH_3_*OC-3). HR-ESI-MS positive mode (m/z): calc. for [M + Na]^+^ = 333.1672, found: [M + Na]^+^ = 333.1679; C_17_H_26_O_5_ (310.39).

**20a**: ^1^H NMR (500 MHz, CDCl_3_) δ 7.45–7.38 (2H, m, aromatics), 7.38–7.30 (2H, m, aromatics), 7.29–7.23 (1H, m, aromatic), 6.77 (1H, d, *J*_1,2_ 12.0 Hz, H-1), 5.58 (1H, dd, *J*_2,3_ 10.0 Hz, H-2), 4.59 (1H, dd, *J*_3,4_ 4.6 Hz, H-3), 3.99–3.91 (1H, m, H-6), 3.57 (3H, s, *CH_3_*OC-4), 3.57–3.49 (3H, m, H-4, H-7_a_, H-7_b_), 3.45 (1H, dd, *J*_4,5_ 3.6, *J*_5,6_ 6.4 Hz, H-5), 3.40 (3H, s, *CH_3_*OC-7), 3.32 (3H, s, *CH_3_*OC-5), 3.23 (3H, s, *CH_3_*OC-3), 3.02 (1H, bs, OH). ^13^C NMR (125 MHz, CDCl_3_) δ 133.9 (C-1), 137.0–126.3 (aromatics), 129.5 (C-2), 84.1 (C-4), 79.6 (C-5), 76.8 (C-3), 73.9 (C-7), 70.4 (C-6), 60.7 (*CH_3_*OC-4), 59.2 (*CH_3_*OC-7), 59.1 (*CH_3_*OC-5), 56.4 (*CH_3_*OC-3). HR-ESI-MS positive mode (m/z): calc. for [M + Na]^+^ = 333.1672, found: [M + Na]^+^ = 333.1669; C_17_H_26_O_5_ (310.39).

#### 4.9.2. (*E*)-1,2-Dideoxy-3,4,5,7-Tetra-*O*-Methyl-1-(4-Trifluoromethylphenyl)-d-*gluco*-Hept-1-Enitol (**19c**)

Isolated from a reaction of tosylhydrazone **17** (0.05 g, 0.13 mmol), (4-trifluoromethly)phenylboronic acid (1.5 equiv., 0.04 g, 0.39 mmol), and K_3_PO_4_ (3 equiv., 0.08 g, 0.39 mmol) according to General procedure I by column chromatography (1:3 EtOAc–hexane) to yield 11 mg white amorphous solid containing **19c** and an unidentified impurity in 3:1 ratio. R_f_: 0.41 (1:2 EtOAc–hexane). ^1^H NMR (500 MHz, CDCl_3_) δ 8.40–7.40 (4H, m, aromatics), 6.68 (1H, d, *J*_1,2_ 16.0 Hz, H-1), 6.29 (1H, dd, *J*_2,3_ 7.7 Hz, H-2), 4.09 (1H, dd, *J*_3,4_ 5.9 Hz, H-3), 4.00–3.91 (1H, m, H-6), 3.60 (3H, s, *CH_3_*OC-4), 3.60–3.53 (3H, m, H-4, H-7_a_, H-7_b_), 3.41 (3H, s, *CH_3_*OC-5), 3.40 (3H, s, *CH_3_*OC-7), 3.39 (3H, s, *CH_3_*OC-3), 3.40–3.36 (1H, m, H-5), 3.07 (1H, bs, OH). ^13^C NMR (90 MHz, CDCl_3_) δ 132.1 (C-1), 140.0–120.5 (aromatics), 129.5 (C-2), 83.5 (C-4), 82.7 (C-3), 79.7 (C-5), 73.6 (C-7), 70.2 (C-6), 60.8 (*CH_3_*OC-4), 59.5 (*CH_3_*OC-7), 59.2 (*CH_3_*OC-5), 57.1 (*CH_3_*OC-3). HR-ESI-MS positive mode (m/z): calc. for [M + Na]^+^ = 401.1546, found: [M + Na]^+^ = 401.1542; C_18_H_25_F_3_O_5_ (378.17).



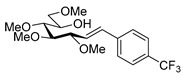



#### 4.9.3. (*E*)-1,2-Dideoxy-1-(4-Fluorophenyl)-3,4,5,7-Tetra-*O*-Methyl-d-*gluco*-Hept-1-Enitol (**19d**) and (*Z*)-1,2-Dideoxy-(4-Fluorophenyl)-3,4,5,7-Tetra-*O*-Methyl-1-d-*gluco*-Hept-1-Enitol (**20d**)

Isolated from a reaction of tosylhydrazone **17** (0.05 g, 0.13 mmol), 4-fluorophenylboronic acid (1.5 equiv., 0.03 g, 0.19 mmol), and K_3_PO_4_ (3 equiv., 0.08 g, 0.39 mmol) according to General procedure I by column chromatography (1:4 EtOAc–hexane) to yield 31 mg white amorphous solid containing **19d** and **20d** in 3:1 ratio. R_f_: 0.11 (1:2 EtOAc–hexane). 



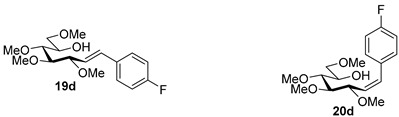



**19d**: ^1^H NMR (400 MHz, CDCl_3_) δ 7.39 (2H, dd, *J* 5.4, 8.7 Hz, aromatics), 7.03 (2H, t, *J* 8.7 Hz, aromatics), 6.60 (1H, d, *J*_1,2_ 16.0 Hz, H-1), 6.09 (1H, dd, *J*_2,3_ 8.1 Hz, H-2), 4.04 (1H, dd, *J*_3,4_ 5.9 Hz, H-3), 4.00–3.91 (1H, m, H-6), 3.59 (3H, s, *CH_3_*OC-4), 3.59–3.49 (3H, m, H-4, H-7_a_, H-7_b_), 3.40 (6H, 2s, *CH_3_*OC-5, *CH_3_*OC-7), 3.38 (1H, dd, *J*_4,5_ 3.1, *J*_5,6_ 7.3 Hz, H-5), 3.36 (3H, s, *CH_3_*OC-3), 3.03 (1H, bs, OH). ^13^C NMR (90 MHz, CDCl_3_) δ 132.7 (C-1), 129.7–110.2 (aromatics), 126.4 (C-2), 83.7 (C-4), 83.2 (C-3), 79.8 (C-5), 73.7 (C-7), 70.2 (C-6), 60.8 (*CH_3_*OC-4), 59.4 (*CH_3_*OC-5), 59.2 (*CH_3_*OC-7), 56.9 (*CH_3_*OC-3). HR-ESI-MS positive mode (m/z): calc. for [M + Na]^+^ = 351.1578, found: [M + Na]^+^ = 351.1579; C_17_H_25_FO_5_ (328.17).

**20d**: ^1^H NMR (500 MHz, CDCl_3_) δ 7.34 (2H, dd, *J* 5.5, 8.5 Hz, aromatics), 7.03 (2H, t, *J* 8.7 Hz, aromatics), 6.72 (1H, d, *J*_1,2_ 11.9 Hz, H-1), 5.66 (1H, dd, *J*_2,3_ 10.1 Hz, H-2), 4.54 (1H, dd, *J*_3,4_ 4.8 Hz, H-3), 3.98–3.91 (1H, m, H-6), 3.57 (3H, s, *CH_3_*OC-4), 3.57–3.50 (3H, m, H-4, H-7_a_, H-7_b_), 3.45 (1H, dd, *J*_4,5_ 3.3, *J*_5,6_ 6.6 Hz, H-5), 3.40 (3H, s, *CH_3_*OC-7), 3.33 (3H, s, *CH_3_*OC-5), 3.22 (3H, s, *CH_3_*OC-3), 3.17 (1H, bs, OH). ^13^C NMR (125 MHz, CDCl_3_) δ 132.7 (C-1), 131.2–114.5 (aromatics), 129.4 (C-2), 84.1 (C-4), 79.6 (C-5), 76.8 (C-3), 73.9 (C-7), 70.4 (C-6), 60.7 (*CH_3_*OC-4), 59.3 (*CH_3_*OC-7), 59.0 (*CH_3_*OC-5), 56.4 (*CH_3_*OC-3). HR-ESI-MS positive mode (m/z): calc. for [M + Na]^+^ = 351.1578, found: [M + Na]^+^ = 351.1579; C_17_H_25_FO_5_ (328.17).

#### 4.9.4. (*E*)-1-(3-Chlorophenyl)-1,2-Dideoxy-3,4,5,7-Tetra-*O*-Methyl-d-*gluco*-Hept-1-Enitol (**19e**) and (*Z*)-1-(3-Chlorophenyl)-1,2-Dideoxy-3,4,5,7-Tetra-*O*-Methyl-d-*gluco*-Hept-1-Enitol (**20e**)

Isolated from a reaction of tosylhydrazone **17** (0.05 g, 0.13 mmol), 3-chorophenylboronic acid (1.5 equiv., 0.03 g, 0.19 mmol), and K_3_PO_4_ (3 equiv., 0.08 g, 0.39 mmol) according to General procedure I by column chromatography (1:2 EtOAc–hexane) to yield 18 mg pale yellow amorphous solid containing **19e** and **20e** in 9:1 ratio. R_f_: 0.13 (1:2 EtOAc–hexane). 



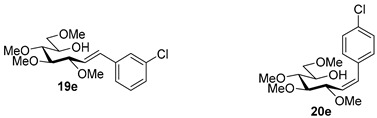



**19e**: ^1^H NMR (400 MHz, CDCl_3_) δ 7.50–7.37 (1H, m, aromatic), 7.36–7.19 (3H, m, aromatics), 6.58 (1H, d, *J*_1,2_ 16.0 Hz, H-1), 6.19 (1H, dd, *J*_2,3_ 7.9 Hz, H-2), 4.06 (1H, dd, *J*_3,4_ 6.2 Hz, H-3), 4.01–3.90 (1H, m, H-6), 3.60 (3H, s, *CH_3_*OC-4), 3.59–3.44 (3H, m, H-4, H-7_a_, H-7_b_), 3.40 (6H, 2s, *CH_3_*OC-5, *CH_3_*OC-7), 3.38 (1H, dd, *J*_4,5_ 2.6, *J*_5,6_ 7.4 Hz, H-5), 3.37 (3H, s, *CH_3_*OC-3), 3.11 (1H, bs, OH). ^13^C NMR (100 MHz, CDCl_3_) δ 132.5 (C-1), 138.8–120.5 (aromatics), 124.9 (C-2), 83.5 (C-4), 82.9 (C-3), 79.7 (C-5), 73.6 (C-7), 70.2 (C-6), 60.8 (*CH_3_*OC-4), 59.4 (*CH_3_*OC-5), 59.2 (*CH_3_*OC-7), 57.0 (*CH_3_*OC-3). HR-ESI-MS positive mode (m/z): calc. for [M + Na]^+^ = 367.1283, found: [M + Na]^+^ = 367.1282; C_17_H_25_ClO_5_ (344.14).

**20e**: ^1^H NMR (400 MHz, CDCl_3_) δ 7.84–6.06 (5H, m, H-1, aromatics), 5.75 (1H, dd, *J*_1,2_ 11.9, *J*_2,3_ 10.01 Hz, H-2), 4.52 (1H, dd, *J*_3,4_ 4.3 Hz, H-3), 4.12–3.71 (1H, m, H-6), 3.57 (3H, s, *CH_3_*OC-4), 3.58–3.42 (4H, m, H-4, H-5, H-7_a_, H-7_b_), 3.40 (3H, s, *CH_3_*OC-7), 3.40–3.23 (3H, m, *CH_3_*OC-5), 3.23 (3H, s, *CH_3_*OC-3), 3.11 (1H, bs, OH). ^13^C NMR (100 MHz, CDCl_3_) δ 132.4 (C-1), 138.8–120.5 (aromatics), 129.3 (C-2), 83.9 (C-4), 83.0 (C-5), 81.4 (C-3), 74.5 (C-7), 73.9 (C-6), 60.4 (*CH_3_*OC-4), 59.3 (*CH_3_*OC-7), 59.1 (*CH_3_*OC-5), 57.0 (*CH_3_*OC-3). HR-ESI-MS positive mode (m/z): calc. for [M + Na]^+^ = 367.1283, found: [M + Na]^+^ = 367.1282; C_17_H_25_ClO_5_ (344.14).

#### 4.9.5. (*E*)-1-(4-Bromophenyl)-1,2-Dideoxy-3,4,5,7-Tetra-*O*-Methyl-d-*gluco*-Hept-1-Enitol (**19f**) and (*Z*)-1-(4-Bromophenyl)-1,2-Dideoxy-3,4,5,7-Tetra-*O*-Methyl-d-*gluco*-Hept-1-Enitol (**20f**)

Isolated from a reaction of tosylhydrazone **17** (0.05 g, 0.13 mmol), 4-bromophenylboronic acid (1.5 equiv., 0.04 g, 0.19 mmol), and K_3_PO_4_ (3 equiv., 0.08 g, 0.39 mmol) according to General procedure I by column chromatography (1:4 EtOAc–hexane) to yield 20 mg white amorphous solid containing **19f** and **20f** in 9:1 ratio. R_f_: 0.10 (1:2 EtOAc–hexane). 



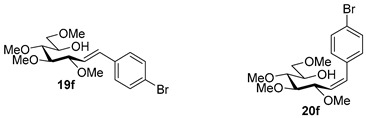



**19f**: ^1^H NMR (400 MHz, CDCl_3_) δ 7.46 (2H, d, *J* 8.5 Hz, aromatics), 7.28 (2H, d, *J* 8.5 Hz, aromatics), 6.57 (1H, d, *J*_1,2_ 16.0 Hz, H-1), 6.18 (1H, dd, *J*_2,3_ 7.9 Hz, H-2), 4.04 (1H, dd, *J*_3,4_ 5.9 Hz, H-3), 3.99–3.90 (1H, m, H-6), 3.59 (3H, s, *CH_3_*OC-4), 3.58–3.50 (3H, m, H-4, H-7_a_, H-7_b_), 3.40 (3H, s, *CH_3_*OC-5), 3.39 (3H, s, *CH_3_*OC-7), 3.37 (3H, s, *CH_3_*OC-3), 3.37 (1H, dd, *J*_4,5_ 2.8, *J*_5,6_ 6.7 Hz, H-5), 3.00 (1H, bs, OH). ^13^C NMR (100 MHz, CDCl_3_) δ 132.4 (C-1), 136.1–117.1 (aromatics), 127.6 (C-2), 83.6 (C-4), 83.0 (C-3), 79.7 (C-5), 73.7 (C-7), 70.2 (C-6), 60.7 (*CH_3_*OC-4), 59.4 (*CH_3_*OC-5), 59.2 (*CH_3_*OC-7), 57.0 (*CH_3_*OC-3). HR-ESI-MS positive mode (m/z): calc. for [M + Na]^+^ = 411.0778, found: [M + Na]^+^ = 411.0777; C_17_H_25_BrO_5_ (389.29).

**20f**: ^1^H NMR (400 MHz, CDCl_3_) δ 7.47 (2H, d, *J* 8.4 Hz, aromatics), 7.25 (2H, d, *J* 8.4 Hz, aromatics), 6.69 (1H, d, *J*_1,2_ 11.9 Hz, H-1), 5.71 (1H, dd, *J*_2,3_ 10.1 Hz, H-2), 4.53 (1H, dd, *J*_3,4_ 4.7 Hz, H-3), 3.99–3.90 (1H, m, H-6), 3.57 (3H, s, *CH_3_*OC-4), 3.56–3.47 (3H, m, H-4, H-7_a_, H-7_b_), 3.45 (1H, dd, *J*_4,5_ 3.3, *J*_5,6_ 6.5 Hz, H-5), 3.40 (3H, s, *CH_3_*OC-7), 3.34 (3H, s, *CH_3_*OC-5), 3.21 (3H, s, *CH_3_*OC-3), 3.00 (1H, bs, OH). ^13^C NMR (100 MHz, CDCl_3_) δ 132.6 (C-1), 136.1–117.1 (aromatics), 130.2 (C-2), 84.0 (C-4), 79.5 (C-5), 76.8 (C-3), 73.8 (C-7), 70.4 (C-6), 60.7 (*CH_3_*OC-4), 59.3 (*CH_3_*OC-7), 59.0 (*CH_3_*OC-5), 56.4 (*CH_3_*OC-3). HR-ESI-MS positive mode (m/z): calc. for [M + Na]^+^ = 411.0778, found: [M + Na]^+^ = 411.0777; C_17_H_25_BrO_5_ (389.29).

#### 4.9.6. (*E*)-1,2-Dideoxy-3,4,5,7-Tetra-*O*-Methyl-1-(4-Methoxyphenyl)-d-*gluco*-Hept-1-Enitol (**19h**) and (*Z*)-1,2-Dideoxy-3,4,5,7-Tetra-*O*-Methyl-1-(4-Methoxyphenyl)-d-*gluco*-Hept-1-Enitol (**20h**)

Isolated from a reaction of tosylhydrazone **17** (0.05 g, 0.13 mmol), 4-methoxyphenylboronic acid (1.5 equiv., 0.03 g, 0.19 mmol), and K_3_PO_4_ (3 equiv., 0.08 g, 0.39 mmol) according to General procedure I by column chromatography (1:2 EtOAc–hexane) to yield 24 mg pale yellow amorphous solid containing **19h** and **20h** in 23:1 ratio. R_f_: 0.13 (1:2 EtOAc–hexane). 



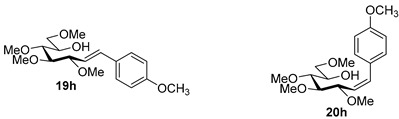



**19h**: ^1^H NMR (400 MHz, CDCl_3_) δ 7.36 (2H, d, *J* 8.7 Hz, aromatics), 6.88 (2H, d, *J* 8.7 Hz, aromatics), 6.57 (1H, d, *J*_1,2_ 16.0 Hz, H-1), 6.01 (1H, dd, *J*_2,3_ 8.3 Hz, H-2), 4.02 (1H, dd, *J*_3,4_ 6.0 Hz, H-3), 3.98–3.91 (1H, m, H-6), 3.82 (3H, s, OCH_3_), 3.60 (3H, s, *CH_3_*OC-4), 3.59–3.53 (2H, m, H-7_a_, H-7_b_), 3.54–3.49 (1H, m, H-4), 3.40 (3H, s, *CH_3_*OC-5), 3.39 (3H, s, *CH_3_*OC-7), 3.38 (1H, dd, *J*_4,5_ 2.9, *J*_5,6_ 6.9 Hz, H-5), 3.35 (3H, s, *CH_3_*OC-3), 3.02 (1H, bs, OH). ^13^C NMR (100 MHz, CDCl_3_) δ 133.6 (C-1), 159.9–112.7 (aromatics), 124.3 (C-2), 83.9 (C-4), 83.6 (C-3), 79.8 (C-5), 73.8 (C-7), 70.3 (C-6), 60.8 (*CH_3_*OC-4), 59.4 (*CH_3_*OC-5), 59.2 (*CH_3_*OC-7), 56.7 (*CH_3_*OC-3), 55.5 (OCH_3_). HR-ESI-MS positive mode (m/z): calc. for [M + Na]^+^ = 363.1778, found: [M + Na]^+^ = 363.1779; C_18_H_28_O_5_ (340.42).

**20h**: ^1^H NMR (400 MHz, CDCl_3_) δ 7.32 (2H, d, *J* 8.7 Hz, aromatics), 6.88 (2H, d, *J* 8.7 Hz, aromatics), 6.69 (1H, d, *J*_1,2_ 12.0 Hz, H-1), 5.56 (1H, dd, *J*_2,3_ 10.0 Hz, H-2), 4.63 (1H, dd, *J*_3,4_ 4.7 Hz, H-3), 3.99–3.90 (1H, m, H-6), 3.82 (3H, s, OCH_3_), 3.58 (3H, s, *CH_3_*OC-4), 3.57–3.49 (3H, m, H-4, H-7_a_, H-7_b_), 3.46 (1H, dd, *J*_4,5_ 3.3, *J*_5,6_ 6.6 Hz, H-5), 3.40 (3H, s, *CH_3_*OC-7), 3.33 (3H, s, *CH_3_*OC-5), 3.23 (3H, s, *CH_3_*OC-3), 3.02 (1H, bs, OH). ^13^C NMR (100 MHz, CDCl_3_) δ 133.3 (C-1), 161.0–112.7 (aromatics), 130.4 (C-2), 84.2 (C-4), 79.6 (C-5), 77.0 (C-3), 73.9 (C-7), 70.4 (C-6), 60.7 (*CH_3_*OC-4), 59.3 (*CH_3_*OC-7), 59.0 (*CH_3_*OC-5), 56.3 (*CH_3_*OC-3), 55.4 (OCH_3_). HR-ESI-MS positive mode (m/z): calc. for [M + Na]^+^ = 363.1778, found: [M + Na]^+^ = 363.1776; C_18_H_28_O_5_ (340.42).

#### 4.9.7. (*E*)-1,2-Dideoxy-3,4,5,7-Tetra-*O*-Methyl-1-(4-Methylphenyl)-d-*gluco*-Hept-1-Enitol (**19i**) and (*Z*)-1,2-Dideoxy-3,4,5,7-Tetra-*O*-Methyl-1-(4-Methylphenyl)-d-*gluco*-Hept-1-Enitol (**20i**)

Isolated from a reaction of tosylhydrazone **17** (0.05 g, 0.13 mmol), 4-methylphenylboronic acid (1.5 equiv., 0.03 g, 0.19 mmol), and K_3_PO_4_ (3 equiv., 0.08 g, 0.39 mmol) according to General procedure I by column chromatography (1:2 EtOAc–hexane) to yield 24 mg pale white amorphous solid containing **19i** and **20i** in 8:1 ratio. R_f_: 0.13 (1:2 EtOAc–hexane), [α]_D_ + 28 (*c* 0.36, CH_2_Cl_2_). 



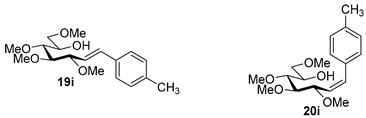



**19i**: ^1^H NMR (400 MHz, CDCl_3_) δ 7.32 (2H, d, *J* 8.1 Hz, aromatics),7.15 (2H, d, *J* 7.9 Hz, aromatics), 6.60 (1H, d, *J*_1,2_ 16.0 Hz, H-1), 6.10 (1H, dd, *J*_2,3_ 8.3 Hz, H-2), 4.03 (1H, dd, *J*_3,4_ 6.0 Hz, H-3), 3.98–3.90 (1H, m, H-6), 3.60 (3H, s, *CH_3_*OC-4), 3.58–3.49 (3H, m, H-4, H-7_a_, H-7_b_), 3.39 (6H, 2s, *CH_3_*OC-5, *CH_3_*OC-7), 3.39–3.36 (1H, m, H-5), 3.35 (3H, s, *CH_3_*OC-3), 3.03 (1H, bs, OH), 2.35 (3H, s, CH_3_). ^13^C NMR (100 MHz, CDCl_3_) δ 134.0 (C-1), 138.3–125.3 (aromatics), 125.5 (C-2), 83.8 (C-4), 83.6 (C-3), 79.8 (C-5), 73.8 (C-7), 70.2 (C-6), 60.8 (*CH_3_*OC-4), 59.4 (*CH_3_*OC-5), 59.2 (*CH_3_*OC-7), 56.7 (*CH_3_*OC-3), 21.3 (CH_3_). HR-ESI-MS positive mode (m/z): calc. for [M + Na]^+^ = 347.1829, found: [M + Na]^+^ = 347.1828; C_18_H_28_O_5_ (324.42).

**20i**: ^1^H NMR (400 MHz, CDCl_3_) δ 7.28–7.23 (4H, m, aromatics), 6.73 (1H, d, *J*_1,2_ 12.1 Hz, H-1), 5.62 (1H, dd, *J*_2,3_ 10.0 Hz, H-2), 4.61 (1H, dd, *J*_3,4_ 4.6 Hz, H-3), 3.98–3.90 (1H, m, H-6), 3.57 (3H, s, *CH_3_*OC-4), 3.58–3.49 (3H, m, H-4, H-7_a_, H-7_b_), 3.46 (1H, dd, *J*_4,5_ 3.4, *J*_5,6_ 6.5 Hz, H-5), 3.40 (3H, s, *CH_3_*OC-7), 3.33 (3H, s, *CH_3_*OC-5), 3.23 (3H, s, *CH_3_*OC-3), 3.03 (1H, bs, OH), 2.36 (3H, s, CH_3_). ^13^C NMR δ 133.7 (C-1), 138.2–125.3 (aromatics), 128.7 (C-2), 84.1 (C-4), 79.6 (C-5), 76.7 (C-3), 73.9 (C-7), 70.5 (C-6), 60.7 (*CH_3_*OC-4), 59.2 (*CH_3_*OC-7), 59.1 (*CH_3_*OC-5), 56.4 (*CH_3_*OC-3), 21.3 (CH_3_). HR-ESI-MS positive mode (m/z): calc. for [M + Na]^+^ = 347.1829, found: [M + Na]^+^ = 347.1828; C_18_H_28_O_5_ (324.42).

### 4.10. 2,6-Anhydro-3,4,5,7-Tetra-O-Methoxymethyl-d-glycero-L-manno-Heptononitrile (2,3,4,6-Tetra-O-Methoxymethyl-β-d-Galactopyranosyl Cyanide) (***23***)

β-d-Galactopyranosyl cyanide **22** (0.10 g, 0.53 mmol) was suspended in dichloromethane (7 mL). The suspension was stirred under nitrogen atmosphere and cooled to 0 °C, and then *N*-diisopropylethylamine (6.4 equiv. / OH, 2.3 mL, 1.75 g, 13.55 mmol) was added, followed by careful addition of chloromethyl methyl ether (10 equiv. / OH, 1.6 mL, 1.70 g, 21.13 mmol), dropwise. The reaction mixture was stirred in the dark at room temperature. When TLC (1:1 EtOAc–hexane) indicated complete consumption of the starting compound (3 day), the mixture was cooled to 0 °C. Saturated aqueous NH_4_Cl solution (1 mL) was added to the reaction mixture. The organic layer was separated, washed with water (1 mL), then the aquous phase was washed with dichloromethane (3 × 3 mL). The combined organic phase was washed with water (1 mL) and dried on anhydrous magnesium sulfate. The solution was concentrated under reduced pressure and purified by column chromatography (1:1 EtOAc–hexane) to yield 163 mg (84%) of **23** as a colourless oil. R_f_: 0.45 (1:1 EtOAc–hexane); [α]_D_ − 40 (*c* 0.29, CHCl_3_). ^1^H NMR (400 MHz, DMSO-*d*_6_) δ 4.86 (1H, d, *J* 6.7 Hz, CH_2_), 4.77 (1H, d, *J* 6.7 Hz, CH_2_), 4.72 (1H, d, *J* 6.6 Hz, CH_2_), 4.71 (1H, d, *J* 6.6 Hz, CH_2_), 4.65–4.58 (3H, m, H-2, CH_2_), 4.57 (2H, s, 2 × CH_2_), 4.00 (1H, dd, *J*_5,6_ 0.6 Hz, H-5), 3.91 (1H, pseudo t, *J*_2,3_ 9.8, *J*_3,4_ 9.6 Hz, H-3), 3.85 (1H, ddd, *J*_6,7a_ 5.9, *J*_6,7b_ 5.9 Hz, H-6), 3.76 (1H, dd, *J*_4,5_ 2.7 Hz, H-4), 3.58 (1H, dd, *J*_7a,7b_ 11.0 Hz, H-7_a_), 3.56 (1H, dd, H-7_b_), 3.37, 3.32, 3.31, 3.26 (12H, 4s, 4 × CH_3_). ^13^C NMR (100 MHz, DMSO-*d*_6_) δ 117.5 (C-1 = CN), 97.0, 95.9, 94.6 (4 × CH_2_), 77.7, 77.2, 72.3, 72.2, 66.6 (C-2–C-6), 66.2 (C-7), 56.1, 55.4, 55.3, 54.8 (4 × CH_3_). HR-ESI-MS positive mode (m/z): calcd. for [M + H]^+^ = 366.1759, found: [M + H]^+^ = 366.1761; C_15_H_27_NO_9_ (365.17).



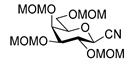



### 4.11. 2,6-Anhydro-3,4,5,7-Tetra-O-Methoxymethyl-d-glycero-L-manno-Heptose Tosylhydrazone (C-(2,3,4,6-Tetra-O-Methoxymethyl-β-d-Galactopyranosyl) Formaldehyde Tosylhydrazone) (***24***) 

Prepared from cyanide **23** (0.10 g, 0.27 mmol) according to General procedure III. Purified by column chromatography (2:1 EtOAc–hexane) to get two unidentified isomers **24-1** and **24-2**.



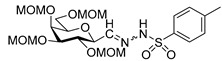



**24-1** yellow oil, 19 mg (13%); R_f_: 0.33 (2:1 EtOAc–hexane). ^1^H NMR (360 MHz, CDCl_3_) δ 9.50 (1H, s, NH), 7.84–7.75 (2H, m, aromatics), 7.33–7.22 (2H, m, aromatics), 4.89 (1H, d, *J* 6.8 Hz, CH_2_), 4.85 (1H, d, *J* 6.5 Hz, CH_2_), 4.79 (1H, d, *J* 6.8 Hz, CH_2_), 4.73–4.59 (4H, m, CH_2_), 4.57 (1H, d, *J* 6.5 Hz, CH_2_), 4.03 (1H, dd, *J*_4,5_ 2.4, *J*_5,6_ 0.6 Hz, H-5), 4.03–3.99 (1H, m, H-2 or H-4), 3.98 (1H, pseudo t, *J*_2,3_ 9.9, *J*_3,4_ 9.9 Hz, H-3), 3.78–3.65 (4H, m, H-2 or H-4, H-6, H-7_a_, H-7_b_), 3.41, 3.39, 3.21 (12H, 4s, 4 × CH_3_), 2.42 (3H, s, CH_3_-Ts). HR-ESI-MS positive mode (m/z): calcd. for [M + H]^+^ = 537.2113, found: [M + H]^+^ = 537.2111; C_22_H_36_N_2_O_11_S (536.20).

**24-2** yellow oil, 96 mg (65%); R_f_: 0.19 (2:1 EtOAc–hexane).^1^H NMR (360 MHz, CDCl_3_) δ 8.25 (1H, s, NH), 7.86–7.73 (2H, m, aromatics), 7.35–7.23 (2H, m, aromatics), 7.05 (1H, d, *J*_1,2_ 4.4 Hz, H-1), 4.87 (1H, d, *J* 6.7 Hz, CH_2_), 4.77 (1H, d, *J* 6.6 Hz, CH_2_), 4.72–4.67 (2H, m, CH_2_), 4.65 (1H, d, *J* 6.7 Hz, CH_2_), 4.60 (2H, s, CH_2_), 4.42 (1H, d, *J* 6.7 Hz, CH_2_), 4.02 (1H, dd, *J*_4,5_ 2.6, *J*_5,6_ 0.6 Hz, H-5), 3.88–3.78 (2H, m) and 3.75–3.55 (4H, m) and 3.46–3.19 (1H, m): (H-2, H-3, H-4, H-6, H-7_a_, H-7_b_), 3.39, 3.32, 3.05 (12H, 4s, 4 × CH_3_), 2.42 (3H, s, CH_3_-Ts). ^13^C NMR (90 MHz, CDCl_3_) δ 146.6 (C-1), 144.8–127.4 (aromatics), 98.2, 97.6, 96.9, 95.7 (4 × CH_2_), 79.1, 78.8, 77.3, 74.6, 72.9 (C-2–C-6), 66.9 (C-7), 56.2, 55.9, 55.6 (4 × CH_3_-Ts). HR-ESI-MS positive mode (m/z): calcd. for [M + H]^+^ = 537.2113, found: [M + H]^+^ = 537.2111; C_22_H_36_N_2_O_11_S (536.20).

### 4.12. Characterization of Anhydro-Heptitol ***25*** and Heptenitols ***26*** and ***27***

#### 4.12.1. 2,6-Anhydro-1-Deoxy-3,4,5,7-Tetra-*O*-Methoxymethyl-1-Phenyl-d-*glycero*-d-*gulo*-Heptitol (**25**)

Isolated from a reaction of tosylhydrazone **24** (0.10 g, 0.19 mmol), phenylboronic acid (1.5 equiv., 0.03 g, 0.28 mmol), and K_3_PO_4_ (3 equiv., 0.12 g, 0.56 mmol) according to General procedure I by column chromatography (1:6 EtOAc–hexane) to yield 7 mg white amorphous solid containing **25** and **28** in 2.6:1 ratio. R_f_: 0.35 (1:2 EtOAc–hexane). ^1^H NMR (400 MHz, CDCl_3_) δ 7.32–7.16 (5H, m, aromatics), 4.98 (1H, d, *J* 6.5 Hz, CH_3_O*CH_2_*OC-3), 4.92 (1H, d, *J* 6.8 Hz, CH_3_O*CH_2_*OC-5), 4.83 (1H, d, *J* 6.8 Hz, CH_3_O*CH_2_*OC-4), 4.76 (H, d, *J* 6.5 Hz, CH_3_O*CH_2_*OC-3), 4.70 (1H, d, *J* 6.8 Hz, CH_3_O*CH_2_*OC-4), 4.69 (1H, d, *J* 7.0 Hz, CH_3_O*CH_2_*OC-5), 4.55 (1H, d, *J* 6.5 Hz, CH_3_O*CH_2_*OC-7), 4.50 (1H, d, *J* 6.5 Hz, CH_3_O*CH_2_*OC-7), 4.05 (1H, dd, *J*_4,5_ 2.0, *J*_5,6_ 0.6 Hz, H-5), 3.76–3.67 (2H, m, H-3, H-4), 3.67 (1H, dd, *J*_6,7a_ 6.3, *J*_7a,7b_ 10.2 Hz, H-7_a_), 3.58 (1H, dd, *J*_6,7b_ 6.5 Hz, H-7_b_), 3.48 (3H, s, *CH_3_*OCH_2_OC-3), 3.50–3.44 (1H, m, H-6), 3.43 (3H, s *CH_3_*OCH_2_OC-4), 3.42 (3H, s *CH_3_*OCH_2_OC-5), 3.42–3.39 (1H, m, H-2), 3.27 (3H, s, *CH_3_*OCH_2_OC-7), 3.23 (1H, dd, *J*_1a,1b_ 14.2, *J*_1a,2_ 1.5 Hz, H-1_a_), 2.77 (1H, dd, *J*_1b,2_ 10.0 Hz, H-1_b_). ^13^C NMR (100 MHz, CDCl_3_) δ 139.7–125.1 (aromatics), 98.9 (CH_3_O*CH_2_*OC-3), 97.5 (CH_3_O*CH_2_*OC-5), 96.9 (CH_3_O*CH_2_*OC-7), 95.4 (CH_3_O*CH_2_*OC-4), 80.9 (C-2), 80.2 (C-4), 77.5 (C-3), 77.2 (C-6), 72.9 (C-5), 66.7 (C-7), 56.7 (*CH_3_*OCH_2_OC-3), 56.1 (*CH_3_*OCH_2_OC-5), 56.0 (*CH_3_*OCH_2_OC-4), 55.5 (*CH_3_*OCH_2_OC-7), 38.1 (C-1). HR-ESI-MS positive mode (m/z): calc. for [M + Na]^+^ = 453.2095, found: [M + Na]^+^ = 453.2093; C_21_H_34_O_9_ (430.49).



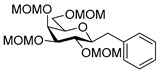



#### 4.12.2. (*E*)-1,2-Dideoxy-3,4,5,7-Tetra-*O*-Methoxymethyl-1-Phenyl-d-*gluco*-Hept-1-Enitol (**26**) and (*Z*)-1,2-Dideoxy-3,4,5,7-Tetra-*O*-Methoxymethyl-1-Phenyl-d-*gluco*-Hept-1-Enitol (**27**)

Isolated from a reaction of tosylhydrazone **24** (0.10 g, 0.19 mmol), phenylboronic acid (1.5 equiv., 0.03 g, 0.28 mmol), and K_3_PO_4_ (3 equiv., 0.12 g, 0.56 mmol) according to General procedure I by column chromatography (1:6 EtOAc–hexane) to yield 19 mg white amorphous solid containing **26** and **27** in 100:1 ratio. R_f_: 0.29 (1:2 EtOAc–hexane), [α]_D_ + 1 (*c* 0.30, CH_2_Cl_2_). 



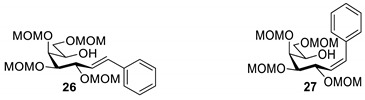



**26**: ^1^H NMR (500 MHz, CDCl_3_) δ 7.39 (2H, d, *J* 8.7 Hz, aromatics), 7.35–7.29 (2H, m, aromatics), 7.29–7.23 (1H, m, aromatic), 6.65 (1H, d, *J*_1,2_ 16.0 Hz, H-1), 6.15 (1H, dd, *J*_2,3_ 8.1 Hz, H-2), 4.86 (1H, d, *J* 6.6 Hz, CH_3_O*CH_2_*OC-4), 4.84 (2H, d, *J* 6.7 Hz, CH_3_O*CH_2_*OC-4, CH_3_O*CH_2_*OC-5), 4.78 (1H, d, *J* 6.7 Hz, CH_3_O*CH_2_*OC-3), 4.71 (1H, d, *J* 6.8 Hz, CH_3_O*CH_2_*OC-5), 4.64 (1H, d, *J* 6.7 Hz, CH_3_O*CH_2_*OC-3), 4.62 (1H, d, *J* 6.5 Hz, CH_3_O*CH_2_*OC-7), 4.60 (1H, d, *J* 6.5 Hz, CH_3_O*CH_2_*OC-7), 4.47 (1H, dd, *J*_3,4_ 5.4 Hz, H-3), 4.20–4.12 (1H, m, H-6), 4.00 (1H, pseudo t, *J*_4,5_ 4.6 Hz, H-4), 3.89 (1H, dd, *J*_5,6_ 2.1 Hz, H-5), 3.66 (1H, dd, *J*_6,7a_ 6.4, *J*_7a,7b_ 10.3 Hz, H-7_a_), 3.64 (1H, dd, *J*_6,7b_ 6.1 Hz, H-7_b_), 3.49 (1H, dd, *J*_6,OH_ 3.9 Hz, OH), 3.46 (3H, s *CH_3_*OCH_2_OC-4), 3.44 (3H, s, *CH_3_*OCH_2_OC-5), 3.41 (3H, s, *CH_3_*OCH_2_OC-3), 3.31 (3H, s, *CH_3_*OCH_2_OC-7). ^13^C NMR (125 MHz, CDCl_3_) δ 134.6 (C-1), 136.4–125.5 (aromatics), 125.9 (C-2), 98.5 (CH_3_O*CH_2_*OC-4), 97.5 (CH_3_O*CH_2_*OC-5), 96.9 (CH_3_O*CH_2_*OC-7), 94.3 (CH_3_O*CH_2_*OC-3), 81.3 (C-4), 77.2 (C-3), 76.9 (C-5), 69.8 (C-6), 69.1 (C-7), 56.4 (*CH_3_*OCH_2_OC-4, *CH_3_*OCH_2_OC-5), 56.0 (*CH_3_*OCH_2_OC-3), 55.4 (*CH_3_*OCH_2_OC-7). HR-ESI-MS positive mode (m/z): calc. for [M + Na]^+^ = 453.2095, found: [M + Na]^+^ = 453.2099; C_21_H_34_O_9_ (430.49).

**27**: ^1^H NMR (500 MHz, CDCl_3_) δ 7.43–7.36 (2H, m, aromatics), 7.35–7.29 (2H, m, aromatics), 7.29–7.23 (1H, m, aromatic), 6.75 (1H, d, *J*_1,2_ 11.4 Hz, H-1), 5.70 (1H, dd, *J*_2,3_ 9.9 Hz, H-2), 4.93–4.22 (11H, m, H-3, H-4, H-5, 4 × CH_3_O*CH_2_*), 4.20–4.12 (1H, m, H-6), 3.96–3.83 (2H, m, H-7_a_, H-7_b_), 3.44, 3.35, 3.34 (12H, 4s, 4 × *CH_3_*OCH_2_). ^13^C NMR (125 MHz, CDCl_3_) δ 133.9 (C-1), 136.4–125.5 (aromatics), 129.2 (C-2), 98.9, 97.6, 97.0, 94.6 (4 × CH_3_O*CH_2_*), 81.6 (C-4), 76.9 (C-5), 71.7 (C-3), 69.5 (C-6), 65.7 (C-7), 56.6, 56.5, 55.9, 55.7 (4 × *CH_3_*OCH_2_). HR-ESI-MS positive mode (m/z): calc. for [M + Na]^+^ = 453.2095, found: [M + Na]^+^ = 453.2099; C_21_H_34_O_9_ (430.49).

## Data Availability

Not applicable.

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
