# Peer review of "Coupling Reactions of Anhydro-Aldose Tosylhydrazones with Boronic Acids"

_molecules, 2022, doi:10.3390/molecules27061795_

Round 1
Reviewer 1 Report
The aim of the present work was the synthesis of C-glycosylmethyl arene derivatives, the hydrolytically stable isosteres of the corresponding aryl O-glycosides. Although relatively simple, these C-glycosides are useful products but quite difficult to obtain as pointed out by the Authors. The carbene-based approach envisaged by the Authors is interesting, however, the desired compounds were formed in low yield and were difficult to purify due to the presence of byproducts.
Variously protected sugar substrates and different reaction conditions were explored without significantly better results. As expected for well-established glycochemistry researchers, the experiments were carefully designed and performed, the observed results were clearly presented, all the new compounds were fully characterized. However, I think that the present manuscript is too limited in scope and therefore it will not attract the interest of the synthetic organic chemistry readership of Molecules since it is more suitable for a carbohydrate chemistry journal.
Minor criticisms:
- Tables 6 and 7 could be deleted because the few negative results can be easily described in the main text.
Author Response
Response to Reviewer 1 Comments
Point 1: However, I think that the present manuscript is too limited in scope and therefore it will not attract the interest of the synthetic organic chemistry readership of Molecules since it is more suitable for a carbohydrate chemistry journal.
Response 1: We thank the referee for the insightful comments and suggestions. Though it is clear that the desired compounds have not been the main products of the reactions, the change in the course of the transformation, namely the formation of the heptenitol type main products as well as the rationale provided to explain this deviation from the established reaction pathway, may be interesting to an audience wider than carbohydrate chemists.
The conclusion has been complemented with the following sentence:
We think that this study also highlights the importance of transformations of high complexity which, though resulting in several products, may lead to a better understanding of their mechanism and may thus inspire further work.
Point 2: Tables 6 and 7 could be deleted because the few negative results can be easily described in the main text.
Response 2: Even if these tables present negative results, due to the use of different substrates as well as reagents, it is believed that omitting them would make it difficult for the reader to understand, so Tables 6 and 7 have not been deleted.
Reviewer 2 Report
In this manuscript, the authors reported metal-free coupling reactions of boronic acids with anhydro-aldose tosylhydrazones with higher complexity. Similar type of strategy already reported by other groups and it is unclear what is new in scientific point of view. The only new thing is coupling with anhydro-aldose tosylhydrazones instead of other. However, I recommend acceptance of this manuscript, still some concerns need to be fixed.
Comments to author
- Citations of previous reports (coupling reactions) must be include in the Scheme 1-2 where required.
- Reaction time should be mention in the tables/Schemes as well.
- The NMR analysis described in the manuscript must be shown as an SI File includes with spectra’s.
- DEPT data for compound 7-10 should be provided.
- Compounds nature whether solid/liquid or crystalline and melting points of newly synthesized should be mention in the exp part.
- I suggest authors to doubly check typos/numbering of all compounds throughout.
Author Response
Response to Reviewer 2 Comments
Point 1: Citations of previous reports (coupling reactions) must be include in the Scheme 1-2 where required.
Response 1: Done.
Point 2: Reaction time should be mention in the tables/Schemes as well.
Response 2: Done.
Point 3: The NMR analysis described in the manuscript must be shown as an SI File includes with spectra’s.
Response 3: Done. An SI with the NMR spectra is uploaded.
Point 4: DEPT data for compound 7-10 should be provided.
Response 4: Exo-glycals 7 and 10 were reported earlier. References have been added to these molecules in the manuscript.
Point 5: Compounds nature whether solid/liquid or crystalline and melting points of newly synthesized should be mention in the exp part.
Response 5: Done.
Point 6: I suggest authors to doubly check typos/numbering of all compounds throughout.
Response 6: Done.
Reviewer 3 Report
This is a well-executed and well written work that addresses an important topic: synthetic approaches to C-glycosides. I have only a couple of minor comments to this manuscript.
Use “2,6-anhydro heptose” rather than “anhydro-aldose”. The former name reflects the range of reagents used in this work more correctly.
In the paper, the authors devote a great deal of effort to optimization of the coupling conditions, with a focus on stoichiometry of the reagents, nature of the carbohydrate protective groups, and the boronic acid aryl substituents. Yet, all the reactions are done in dioxane at 110 C. Why were the solvent and temperature kept unchanged? Did the authors test other solvents or solvent-free conditions, other temperatures, or microwave irradiation? A short discussion on this subject would be of interest to the readers.
Page 4, line 114: “the Li-salt reactions appeared less effective”. Please, explain, less effective than what and by which criteria. Table 1 indicates that the highest yield of desired 2a was achieved with the Li-salt though.
Author Response
Response to Reviewer 3 Comments
Point 1: Use “2,6-anhydro heptose” rather than “anhydro-aldose”. The former name reflects the range of reagents used in this work more correctly.
Response 1: The reviewer is correct, nevertheless, in our previous works we used the “anhydro-aldose” term, therefore, would prefer keeping this name.
Point 2: In the paper, the authors devote a great deal of effort to optimization of the coupling conditions, with a focus on stoichiometry of the reagents, nature of the carbohydrate protective groups, and the boronic acid aryl substituents. Yet, all the reactions are done in dioxane at 110 C. Why were the solvent and temperature kept unchanged? Did the authors test other solvents or solvent-free conditions, other temperatures, or microwave irradiation? A short discussion on this subject would be of interest to the readers.
Response 2: In previous works (refs. [8,10]) we studied the solvent effects on the carbene formation step to show the best performance in refluxing dioxane. Nevertheless, some new experiments have been performed in 3 other solvents (acetonitrile, fluorobenzene and toluene), commonly used in the coupling reactions. The results are shown in Table 1 (entries 6-8) of the manuscript and discussed in the text.
Point 3: Page 4, line 114: “the Li-salt reactions appeared less effective”. Please, explain, less effective than what and by which criteria. Table 1 indicates that the highest yield of desired 2a was achieved with the Li-salt though.
Response 3: We have checked our previous results (Table 1 entry 12 in the original manuscript) with the Li salt and noticed that yields of C-glycosyl derivative 2a were incorrectly presented. C-Glycosyl derivative 2a and exo-glycal 5 gave an inseparable mixture to give products in 7 and 15% yields, respectively. We've corrected our regrettable mistake. So this reaction with the Li salt resulted in heptenitols 3a and 4a as main products similar to the former reactions.
Round 2
Reviewer 1 Report
The manuscript has been modified as requested by the Referees, therefore, the revised version can be accepted without further changes.
Reviewer 2 Report
The suggested changes has been done by authors reasonable and revised version and suitable for acceptance.